
# A high-resolution physical-biogeochemical model for marine resource applications in the Northwest Atlantic (MOM6-COBALT-NWA12 v1.0)

Andrew C. Ross[1], Charles A. Stock[1], Alistair Adcroft[1], Enrique Curchitser[2], Robert Hallberg[1], Matthew J. Harrison[1], Katherine Hedstrom[3], Niki Zadeh[1], Michael Alexander[4], Wenhao Chen[5], Elizabeth J. Drenkard[1], Hubert du Pontavice[5], Raphael Dussin[1,6], Fabian Gomez[7,8], Jasmin G. John[8], Dujuan Kang[9,2], Diane Lavoie[10], Laure Resplandy[11,12], Alizée Roobaert[13], Vincent Saba[14], Sang-Ik Shin[4], Samantha Siedlecki[15], and James Simkins[2]

[1]NOAA/OAR/Geophysical Fluid Dynamics Laboratory, Princeton, NJ, USA
[2]Department of Environmental Sciences, Rutgers University, New Brunswick, NJ, USA
[3]College of Fisheries and Ocean Sciences, University of Alaska Fairbanks, Fairbanks, AK, USA
[4]NOAA Physical Sciences Laboratory, Boulder, CO, USA
[5]Atmospheric and Oceanic Sciences Program, Princeton University, Princeton, NJ, USA
[6]University Corporation for Atmospheric Research, Boulder, CO, USA
[7]Northern Gulf Institute, Mississippi State University, Starkville, MS, USA
[8]NOAA/OAR/Atlantic Oceanographic and Meteorological Laboratory, Miami, FL, USA
[9]School of Oceanography, Shanghai Jiao Tong University, Shanghai, China
[10]Fisheries and Oceans Canada, Maurice Lamontagne Institute, Mont-Joli, Québec, Canada
[11]Department of Geosciences, Princeton University, Princeton, NJ, USA
[12]High Meadows Environmental Institute, Princeton University, Princeton, NJ, USA
[13]Department of Geosciences, Environment & Society-BGEOSYS, Université Libre de Bruxelles, Brussels, CP160/02, Belgium
[14]NOAA Northeast Fisheries Science Center, Geophysical Fluid Dynamics Laboratory, Princeton, NJ, USA
[15]Department of Marine Sciences, University of Connecticut, Groton, CT, USA

**Correspondence:** Andrew C. Ross (andrew.c.ross@noaa.gov)

**Abstract.** We present the development and evaluation of MOM6-COBALT-NWA12 version 1.0, a 1/12° model of ocean dynamics and biogeochemistry in the Northwest Atlantic Ocean. This model is built using the new regional capabilities in the MOM6 ocean model and is coupled with the COBALT biogeochemical model and SIS2 sea ice model. Our goal was to develop a model to provide information to support living marine resource applications across management time horizons from seasons to decades. To do this, we struck a balance between a broad, coastwide domain to simulate basin-scale variability and capture cross-boundary issues expected under climate change, high enough spatial resolution accurately simulate features like the Gulf Stream separation and advection of water masses through finer-scale coastal features, and the computational economy required to run the long simulations of multiple ensemble members that are needed to quantify prediction uncertainties and produce actionable information. We assess whether MOM6-COBALT-NWA12 is capable of supporting the intended applications by evaluating the model with three categories of metrics: basin-wide indicators of the model's performance, indicators of coastal ecosystem variability and the regional ocean features that drive it, and model run times and computational efficiency. Overall,





both the basin-wide and regional ecosystem-relevant indicators are simulated well by the model. Where notable model biases and errors are present in both types of indicators, they are mainly consistent with the challenges of accurately simulating the Gulf Stream separation, path, and variability: for example, the coastal ocean and shelf north of Cape Hatteras is too warm and
salty and has minor biogeochemical biases. During model development, we identified a few model parameters that exerted a notable influence on the model solution, including the horizontal viscosity, mixed layer restratification, and tidal self-attraction and loading, which we discuss briefly. The computational performance of the model is adequate to support running numerous long simulations, even with the inclusion of coupled biogeochemistry with 40 additional tracers. Overall, these results show that this first version of a regional MOM6 model for the Northwest Atlantic Ocean is capable of efficiently and accurately sim-
ulating historical basin-wide and regional mean conditions and variability, laying the groundwork for future studies to analyze this variability in detail, develop and improve parameterizations and model components to better capture local ocean features, and develop predictions and projections of future conditions to support living marine resource applications across time scales.

## 1  Introduction

Over the last few decades, the Northwest Atlantic Ocean has experienced prominent variability and sharp trends driven by
climate change and other anthropogenic factors, shifting currents, and basin-scale modes of climate variability. Sea surface temperatures in the Gulf of Maine and Scotian Shelf and Slope regions, located within the Northeast U.S. Large Marine Ecosystem (LME), warmed faster than the vast majority of the global ocean in the last two decades (Pershing et al., 2015; Seidov et al., 2021). In addition to being driven by a warming atmosphere caused by increasing greenhouse gas concentrations, some of this warming occurred abruptly following shifts in the Gulf Stream path (Friedland et al., 2020a, b). The destabiliza-
tion point of the Gulf Stream has recently moved westward, closer to where it separates from the continental shelf (Andres, 2016), and more frequent intrusions of warm and saline water onto the shelf (Gawarkiewicz et al., 2022) and eddy shedding (Gangopadhyay et al., 2019, 2020) have been observed. Northward shifts of the Gulf Stream have cut off the cool, southward Labrador Current and amplified warming in the region (Gonçalves Neto et al., 2021; Seidov et al., 2021). Sea surface temperatures have also been observed to be warming faster than the global average in the Gulf of Mexico (Wang et al., 2023) and
Caribbean Sea (Glenn et al., 2015). Rising temperatures and changes in ocean dynamics have contributed to a rapid increase in sea level and coastal flooding risk along most of the U.S. East Coast (Ezer and Atkinson, 2014).

Basin-scale modes of climate variability have also contributed to some of the recent variability in the Northwest Atlantic Ocean, although the precise connections are sometimes tenuous. The Atlantic Multidecadal Variability (AMV) or Oscillation (AMO) has recently been in a positive phase that is associated with basin-scale warming (Ting et al., 2009). Recent weakening
Atlantic Meridional Overturning Circulation has amplified warming along the U.S. East Coast and cooled the subpolar gyre (Caesar et al., 2018; Jackson et al., 2022). Fluctuations of the North Atlantic Oscillation (NAO) have been linked to changes in the Labrador Current and Gulf Stream and downstream variability of temperature and salinity in the Northeast U.S. LME between approximately 1 and 4 years later (Grodsky et al., 2017; Mountain, 2012; Xu et al., 2015). Remote climate teleconnections have also been identified, including a link between the Pacific Decadal Oscillation (PDO) and sea surface temperatures



along the Northeast U.S. shelf (Chen and Kwon, 2018). El Niño events have been linked to anomalously fresh and cool conditions in the Gulf of Mexico and warm surface water in the tropical North Atlantic (Alexander and Scott, 2002; Gomez et al., 2019).

The physical variations and trends described above have been accompanied by biogeochemical changes. The same accumulation of carbon dioxide in the atmosphere contributing to ocean warming has acidified the ocean globally, though circulation
changes and warming water are delaying these impacts in some regions of the Northwest Atlantic (Salisbury and Jönsson, 2018; Balch et al., 2022). The near disappearance of oxygenated water from the Labrador Current has increased hypoxia in the Gulf of St. Lawrence and the surrounding shelf (Petrie and Yeats, 2000; Gilbert et al., 2005; Claret et al., 2018; Jutras et al., 2020, 2023). These changes, combined with concomitant shifts in stratification, have shifted seasonal patterns of plankton productivity and zooplankton assemblages (e.g., Balch et al., 2022; Friedland et al., 2023; Morse et al., 2017) and altered harmful
algal blooms (Clark et al., 2019; Heil and Muni-Morgan, 2021; Townsend et al., 2014). At the terrestrial interface, changes in land use and precipitation patterns have altered the delivery of nutrients and alkalinity to coastal waters (Rabalais et al., 1996; Stets et al., 2014; Turner, 2021), shifting hypoxia and coastal acidification patterns (Cai et al., 2011; Gomez et al., 2021; Rabalais et al., 2007).

The physical and biogeochemical changes in the Northwest Atlantic have been associated with pronounced shifts in the
distribution, phenology, and productivity of living marine resources (LMRs) and led to significant ecosystem, socio-economic, and public health consequences. In the Northeast U.S., most fish habitats have shifted to the north (Bell et al., 2015; Pinsky and Fogarty, 2012; Lucey and Nye, 2010; Nye et al., 2009) or offshore and to deeper water (Kleisner et al., 2016; Mazur et al., 2020; Nye et al., 2009) as the ocean has warmed, although Bell et al. (2015) found that not all shifts can be directly connected to temperature. Many of these shifts have occurred across historical management, political, and regional ocean modeling
boundaries, emphasizing the need for a broad, coastwide perspective. Species distribution changes are likely to continue as climate change progresses, both in the Northwest Atlantic and globally, leading to geopolitical conflicts and management challenges as species cross the boundaries of Exclusive Economic Zones (Pinsky et al., 2018) and fishers struggle to keep up with the changes (Pinsky and Fogarty, 2012). Warming water in the Northwest Atlantic also increased cod mortality, resulting in a collapse of the cod fishery (Pershing et al., 2015), and decreased the abundance of *Calanus finmarchicus*, leading to an
extinction-level threat to the North Atlantic Right Whale that feeds on it (Meyer-Gutbrod et al., 2021), and is likely to cause an expansion of HABs in the Gulf of St. Lawrence (Boivin-Rioux et al., 2022).

Shorter-term climate variability can also have substantial impacts, such as the link between a positive phase of the NAO with lower fish catches and decreased fishing employment and wages in New England (Oremus, 2019). River discharge and upwelling have produced frequent co-occurrences of hypoxia and harmful algal blooms (HABs) on the West Florida Shelf
(Turley et al., 2022). The compound stresses of marine heatwaves, ocean warming, and acidification pose an increasing threat to coral reefs in the Caribbean Sea and other parts of the subtropical North Atlantic, with effects further exacerbated by chronic water quality challenges (Hoegh-Guldberg et al., 2007; Donovan et al., 2021; Leggat et al., 2019), and also threaten bivalves in the coastal North Atlantic (Griffith and Gobler, 2017; Waldbusser et al., 2015).





Numerous studies suggest that it is possible to anticipate some of the physical and biogeochemical changes in the Northwest
Atlantic through dynamical or statistical forecasts (e.g., Stock et al., 2015; Tommasi et al., 2017a; Chen et al., 2021). Further-
more, a growing number of cases studies suggest that such predictions can improve marine resource management decisions and
contribute to resilient marine ecosystems and coastal communities (Tommasi et al., 2017c). In the Northwest Atlantic, Mills
et al. (2017) found that observed early spring temperatures anomalies in the Gulf of Maine could skillfully predict the start date
of high lobster landings, with a potential to moderate supply chain disruptions associated with anomalous ocean conditions
(Mills et al., 2013). Miller et al. (2016) and du Pontavice et al. (2022) showed that accounting for the effect of the Cold Pool (a
seasonally formed cold water mass at the bottom of the Northeast U.S. continental shelf) on yellowtail flounder recruitment in
a stock assessment model can improve the predictive skill of recruitment and spawning stock biomass. At decadal timescales,
Tommasi et al. (2017a) found that skillful probabilistic of decadal-scale temperature anomalies was possible if viewed relative
to the 50 year baseline of trawl survey data, suggesting the potential to anticipate species range shifts on the decadal timescales
of capital investments by fishers (Tommasi et al., 2017b). While such outcomes are promising, the uptake of ocean predictions
into LMR management has been slowed in part by limited availability of skillful high-resolution predictions across the range
of marine resource-relevant physical and biogeochemical variables with reliable estimates of prediction uncertainty.

Improved understanding of the drivers of historical trends and the capability to predict and project LMR-relevant future
changes in the Northwest Atlantic requires modeling systems that have both enough spatial resolution and complexity to
resolve physical and biogeochemical processes across scales and enough computational efficiency to run ensemble simulations
that represent uncertainty about the future. In the Northwest Atlantic, past results suggest that ocean resolution on the order
of 1/10° or higher is required to accurately simulate features of the western boundary current, including the separation and
downstream path of the Gulf Stream (Chassignet and Marshall, 2008; Chassignet and Xu, 2017, 2021) and the dynamics of
Loop Current eddies (Oey et al., 2013), and smaller but ecologically critical local ocean features, such as the narrow Northeast
Channel in the Gulf of Maine that is a deep passageway for water from the slope (Saba et al., 2016) and the coastal hypoxia
zone on the Louisiana-Texas Shelf (Fennel et al., 2013). Future projections of climate for the region from lower resolution
models that fail to simulate these features can differ substantially from higher resolution models that do (Liu et al., 2012;
Drenkard et al., 2021; Li et al., 2022). The computational cost of the resolution and complexity needed to simulate such
features conflicts with the desire to temper the computational demands of multiple ensemble members of lengthy simulations
required to project the range of ocean futures across management-relevant time horizons from seasons to centuries (Drenkard
et al., 2021). For example, current-generation seasonal ocean prediction systems typically rely on low ocean resolution to
balance the computational demands of running decades of retrospective forecasts with multiple ensemble members, and these
systems have markedly lower forecast skill for the U.S. East Coast compared to other ocean and coastal regions (Stock et al.,
2015; Hervieux et al., 2017; Shin and Newman, 2021).

Regional ocean and ecosystem modeling systems can bring the benefits of high model grid resolution and complexity to an
area of interest while maintaining the computational feasibility of running decadal to centennial scale simulations with many
ensemble members. Regional models also allow for region-specific optimization of model parameters and the inclusion of
regionally important processes that are too computationally costly or not represented in global model simulations. For example,





current-generation global ocean models typically simulate tides and their effects only implicitly through parameterizations

(Holt et al., 2017), while a regional model that explicitly includes tides may be able to represent processes that are known to be important in the Northwest Atlantic, such as spring-neap variability of temperature and tidal pumping of nutrients in Georges Bank (Bisagni and Sano, 1993; Hu et al., 2008) and eddy activity in the Gulf Stream region (Chassignet and Xu, 2021). For these reasons, regional ocean models will continue to be an important tool even as the resolution and complexity of global models improves with increasing processor speed and supercomputing power (Drenkard et al., 2021). However, although

regional ocean models offer a number of advantages, they also present new challenges. The ocean boundary conditions in a regional model, for example, often exert a substantial influence over the interior solution (e.g., Ghantous et al., 2020), yet specification of these boundary conditions is an ill-posed problem (Bennett and Kloeden, 1981; Marchesiello et al., 2001; Oliger and Sundstrom, 1978). In addition, while regional domains temper the computational constraints of global simulations, they do not eliminate them. A balance must still be struck between sufficient resolution and computational economy if ensemble

predictions spanning the range of potential ocean futures are to be generated.

In this paper, we describe the development and evaluation of a baseline 1/12° regional ocean and biogeochemical model for retrospective and predictive applications in the Northwest Atlantic Ocean: MOM6-COBALT-NWA12 v1.0 (Fig. 1a). This model is derived from the Geophysical Fluid Dynamics Laboratory's (GFDL's) global ocean (MOM6), sea-ice (SIS2), and ocean biogeochemistry (COBALT) models (Adcroft et al., 2019; Stock et al., 2020) and combines the features of the global

versions of these models, including computational efficiency and stability, with newly developed open boundary conditions and regional modeling capabilities to deliver a feature-rich yet computationally tractable regional ocean physical-ecosystem model. The model is intended to support marine resource applications across management time horizons from weeks to seasons to multiple decades and has a "coastwide" extent to address the prominent cross-boundary issues expected under climate change. The model domain also extends considerably into the North Atlantic basin to smoothly connect basin-scale and coastal drivers

of ecosystem change.

Confidence in the intended applications is linked to the model's capacity to accurately simulate past observed responses across these scales. Thus, in the sections that follow, we detail the development and configuration of the model components for a historical simulation covering 1993 to 2019, evaluate the ability of the model to reproduce the historical means and variability of metrics with ecosystem relevance, and discuss several notable sensitivities that we identified during model development.

Finally, we emphasize that the configuration herein is intended as an open development baseline supporting sustained model improvement. We thus conclude with a discussion of model strengths and limitations, with an eye toward future improvements, with a long-term goal of supporting climate-informed decisions across LMR management and decision-making time horizons from seasons to multiple decades.



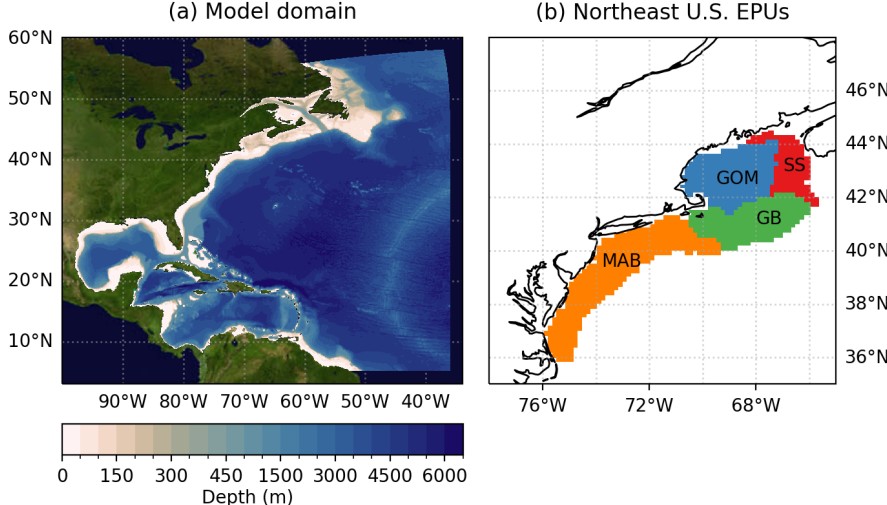

**Figure 1.** (a) Model domain and bathymetry. The color scale spacing is more detailed in the first 500 m to show shallow bathymetric features on the shelves. (b) Northeast U.S. ecological production units (EPUs) used for some model evaluation metrics: Scotian Shelf (SS), Gulf of Maine (GOM), Georges Bank (GB), and Mid-Atlantic Bight (MAB).

## 2 Methods

MOM6-COBALT-NWA12 is comprised of coupled model components for ocean physics, ocean biogeochemistry, and sea ice. In this section, we detail the development and configuration of the regional model components and describe the configuration and evaluation of a reanalysis-forced simulation to determine whether the model is fit for purpose.

### 2.1 Physical ocean model configuration

A summary of the main configuration choices used in the MOM6 component of the model is presented in Table 1. Horizontally,
the model is based on an Arakawa C grid (Arakawa and Lamb, 1977) with 775x845 tracer points. The horizontal grid and bathymetry, shown in Figure 1a, are extracted from the larger North Atlantic basin model of Xu et al. (2010), with open boundaries along the south, east, and north edges. Compared to a previous model based on the Regional Ocean Modeling System that was developed by several coauthors of this study (Kang and Curchitser, 2013, 2015) and applied for climate and ecosystem downscaling simulations (Alexander et al., 2020; Baumann et al., 2022; Clark et al., 2022), the NWA12 model
domain is expanded to include the critical Grand Banks region in the northeast, cover the full Caribbean Sea including the coasts of Puerto Rico and the Virgin Islands, and generally place the open boundaries farther from coastal regions of interest. The resolution is 1/12° throughout most of the domain, and the zonal distance between grid points varies primarily with latitude from approximately 9 km at the southern boundary to 5 km at the northern boundary. Vertically, the model uses a z* coordinate (a height coordinate that is rescaled with the free surface; Adcroft and Campin, 2004) with 75 layers, identical to the z*-only





configuration of the global OM4 model in Adcroft et al. (2019, hereafter A19) but different from the hybrid $z^*$-isopycnal configurations also developed in A19 and employed in GFDL's global climate and earth system models. The vertical resolution is finest near the surface, where the layer thickness is 2 m, and the layer thickness gradually increases with depth to a maximum thickness of 250 m above the deepest model depth of 6500 m. The shallowest bathymetry in the model is 10 m. The model is integrated forward in time with a split explicit method (Hallberg, 1997; Hallberg and Adcroft, 2009). The baroclinic time

step is 600 seconds, and the time-varying barotropic time step is set to the largest integer fraction of the baroclinic time step that is less than 90% of the maximum stable time step. To increase computational efficiency, we used MOM6's sub-cycled time-step capabilities to update the thermodynamics and biogeochemistry (BGC) on a slower 1800 second time step, and the atmospheric forcing data and external biogeochemical boundary data are updated hourly. The computational efficiency gained through these choices will be discussed in the Results.

The ocean model subgrid-scale parameterizations are adapted from the 1/4° global model of A19, with updates and modifications to account for the increased horizontal resolution and other differences in MOM6-NWA12. The planetary boundary layer is parameterized using the same Reichl and Hallberg (2018) energetic planetary boundary layer scheme; however, the parameterization in NWA12 includes the updates by Reichl and Li (2019) to account for enhanced mixing by Langmuir turbulence. Since our resolution resolves the first baroclinic deformation radius across the majority of the model domain, except on the

shelves (Hallberg, 2013), and thus captures a considerable fraction of the mesoscale dynamics, we did not include a mesoscale eddy mixing parameterization (nor did the 1/4° configuration in A19). Restratification of the mixed layer by submesoscale eddies is parameterized using the scheme based on Fox-Kemper et al. (2011). As in A19, we found a strong sensitivity of the simulated mixed layer depths to the submesoscale front length parameter in this scheme, and we increased the front length from 500 m in A19 (which is on the lower end of values typically used; Bodner et al., 2023) to 1500 m. This increased front

length decreases restratification arising from the parameterization. Like A19, the horizontal biharmonic viscosity is formulated as the maximum of either a Smagorinsky viscosity or a fixed viscosity of the form $u_4 \Delta_x^3$ where $\Delta_x$ is the local grid spacing (Griffies and Hallberg, 2000). NWA12 uses $u_4 = 1$ cm s$^{-1}$, the same as A19, with a Smagorinsky coefficient of 0.015, which is reduced from 0.06 in A19. Mixing by shear-driven turbulence is modeled using the Jackson et al. (2008) parameterization as in A19.

There are a few other differences from A19. Most notably, MOM6-COBALT-NWA12 includes explicit tidal dynamics forced by both the boundary and the astronomical tidal potential. We thus removed the unresolved tidal velocity from the formulation of the bottom drag, and we did not include parameterized mixing or the additional tracer diffusivity that A19 used to account for internal tides. Background vertical viscosity was reduced to its molecular value. We also neglected the parameterized conversion of energy dissipated in the bottom boundary layer to diapycnal mixing by reducing the parameter for the efficiency

of this process from 0.20 in A19 to 0.0. When this parameterization was enabled, the model tended to produce exaggerated mixing at the bottom of many coastal areas, which prevented the development of realistic bottom hypoxia in the coastal Gulf of Mexico. The efficiency is considered poorly constrained (generally within a range of 0.0–0.2) and tunable (Gregg et al., 2018; Legg et al., 2006), although eliminating this parameterization of physical mixing may partially compensate for numerically induced mixing in the $z^*$ coordinate model (Griffies et al., 2000).





**Table 1.** Major parameters and associated values used in the physical ocean (MOM6) component of the model and relevant references describing the parameter or the choice of value.

| Parameter | Value | Reference |
|---|---|---|
| Vertical coordinate | 75 layer $z^*$ | Adcroft et al. (2019) |
| Baroclinic time step | 600 s | |
| Thermodynamics and BGC time step | 1800 s | |
| Planetary boundary layer parameterization | ePBL | Reichl and Hallberg (2018) |
| Submesoscale eddy front length | 1500 m | Fox-Kemper et al. (2011); Bodner et al. (2023) |
| Biharmonic viscosity | Maximum of Smagorinsky and resolution-dependent viscosities | Griffies and Hallberg (2000) |
| Smagorinsky coefficient | 0.015 | |
| Resolution-dependent | $0.01\Delta_x^3$ m$^4$ s$^{-1}$ | Adcroft et al. (2019) |
| Bottom boundary layer mixing efficiency | 0.0 | |
| Background kinematic viscosity | $1.0 \times 10^{-6}$ m$^2$ s$^{-1}$ | |
| Background diapycnal diffusivity | $1.0 \times 10^{-6}$ m$^2$ s$^{-1}$ | |
| Boundary conditions | | |
| Sea level and barotropic velocity | Flather scheme | Flather (1976) |
| Baroclinic velocity | Radiation and nudging scheme (3 day inflow, 360 day outflow timescales) | Marchesiello et al. (2001); Orlanski (1976) |
| Tracers | Reservoirs with 9 km length scales | |
| Tidal SAL coefficient | 0.01 | Irazoqui Apecechea et al. (2017); Stepanov and Hughes (2004) |
| Opacity scheme | 3-band with chlorophyll | Manizza (2005) |

As in A19, MOM6-COBALT-NWA12 simulates sea ice and its interaction with the ocean using a coupled sea ice model, Sea Ice Simulator version 2 (SIS2). We refer the reader to A19 for a description of the SIS2 model and configuration; the configuration used in MOM6-COBALT-NWA12 is identical except we have reduced the ice dynamics time step to 600 seconds to match the ocean baroclinic time step. SIS2 does not yet support open boundary conditions for sea ice.

### 2.1.1 Physical ocean model forcing

Boundary conditions for combined tidal and subtidal sea level and barotropic velocity are set using a Flather (1976) radiation boundary condition. Baroclinic flow at the boundary is set using the radiation scheme of Orlanski (1976) combined with nudging towards external forcing data following Marchesiello et al. (2001). Boundary normal and tangential velocities are





strongly nudged towards the forcing data with a 3 day time scale for flow entering the model and weakly nudged with a 360 day timescale for outgoing flow. Temperature and salinity boundary conditions are set using a reservoir scheme that gradually

adjusts boundary data towards interior values on outflow and exterior values on inflow, which allows the tracer boundary conditions to retain a memory of the properties of flow that exits and reenters the domain. The reservoir length scale is set to 9 km for both inflow and outflow, which is approximately equal to a 1–10 day time scale for a 10–1 cm s$^{-1}$ flow and allows the boundary to adjust to changes on weather time scales. The development of and sensitivity to this novel reservoir scheme are being addressed in a separate publication. It is important to note that for all prognostic model variables there is no nudging

to observed data within the model domain—the model does not use a "sponge layer" near the boundary or apply restoring of properties like surface salinity. Attaining a reliable solution without using these features was a deliberate emphasis of model development and facilitates use of the model in downscaled climate projections and other scenarios where accurate internal data to restore to may not be available.

In the reanalysis-driven hindcast simulation presented in this paper, external boundary data for temperature, salinity, and

subtidal velocity and sea level were specified using daily averages from the GLORYS12 v1 ocean reanalysis (Lellouche et al., 2021). This reanalysis provides high resolution (1/12°) daily data, and has been found to be one of the better performing ocean reanalyses in coastal areas including the Northeast U.S. (Castillo-Trujillo et al., 2022) and California Current System (Amaya et al., 2023). Tidal variations in sea level and velocity were superimposed on the subtidal boundary data using tidal harmonics from the TPXO9 v1 dataset (Egbert and Erofeeva, 2002). Four semidiurnal constituents (M2, S2, N2, and K2), four

diurnal constituents (K1, O1, P1, Q1) and two long-period constituents (Mm and Mf) were included in the boundary forcing. Modulation by the 18.6-year nodal cycle was included by calculating correction factors for the amplitude and phase of each constituent at yearly intervals. Astronomical tidal forcing from the same ten constituents was also included throughout the domain as a body force in the momentum equations. The effects of self-attraction and loading were also included using the scalar approximation (Accad and Pekeris, 1978) with a coefficient of 0.01.

Freshwater discharge from rivers was sourced from the gridded daily GloFAS reanalysis version 3.1 (Alfieri et al., 2020). River discharge was mapped to the MOM6 grid by using the local drainage direction map to identify outlet points adjacent to the coast, and any chains of outlet points adjacent to these coastal outlets, and mapping the streamflow at these outlet points to the nearest MOM6 coastal ocean grid cell. River discharge was added at the surface at the discharge grid cells, and an additional source of turbulent kinetic energy was added at discharge points to vertically mix the water column up to 5 m deep.

We found that the GloFAS product overestimated the streamflow in the Mississippi River, so we applied a bias correction using a linear regression between the GloFAS Mississippi River discharge and the USGS gauge at Belle Chasse, LA. We evenly divided this adjusted flow into two points that discharged on the western side of the river delta and one point on the eastern side, which is roughly consistent with the real partitioning of discharge from the delta (Dagg and Breed, 2003; Dinnel and Wiseman, 1986). River discharge from the Atchafalaya River did not appear to need adjustment. We also manually adjusted

the location of discharge from the Susquehanna River to ensure that it entered the model at the correct location in Chesapeake Bay.





Momentum, heat, freshwater, and radiation fluxes between the ocean and atmosphere were calculated using the hourly ERA5 atmospheric reanalysis (Hersbach et al., 2020) and the Large and Yeager (2004) bulk algorithm, with the inclusion of the Large and Yeager (2004) adjustment for the temperature and humidity reference height of 2 m in the ERA5 data. Chlorophyll

predicted by the coupled COBALT biogeochemical component of the model, presented next, was used to specify the vertical profile of the absorption of shortwave radiation following the Manizza (2005) scheme.

## 2.2 Biogeochemical model configuration

The MOM6 physical ocean component is coupled with the Carbon, Ocean Biogeochemistry and Lower Trophics (COBALT) biogeochemical model (Stock et al., 2014, 2020), with a number of enhancements intended to improve the robustness of the

biogeochemical model across the spectrum of coastal to open ocean environments included in the NWA12 domain. For the plankton food web, we drew from Van Oostende et al. (2018) to add a fourth phytoplankton group to better represent spring bloom diatoms that can be particularly prominent in high productivity coastal regions. The phytoplankton parameters enlisted in this run are summarized in Table A1. Direct phytoplankton sinking was added to complement phytoplankton aggregation losses under poor growth conditions and better capture the absence of large diatoms in the subtropical gyres. Upon reaching

the bottom, these slow sinking particles were assumed to settle into a nepheloid layer and be available for resuspension if the bottom was shallower than twice the depth of the actively mixed layer, and were otherwise remineralized. Other aspects of particle sinking and remineralization are as described in Stock et al. (2020).

Phytoplankton in very deep mixed layers (those exceeding 3 e-folding depths) were assumed to photoacclimate to mean light levels over the first 3 e-folding depths, while those in shallow mixed layers were assumed to photoacclimate to mean light

levels over the mixed layer. This is consistent with results of optimality models suggesting that phytoplankton in deep mixed layers must adapt to elevated light conditions closer to the surface (Talmy et al., 2013), and was found to improve the model's representation of offshore bloom timing and magnitude in the NWA12 domain.

Increased flexibility was added to the zooplankton grazing kernels to better represent the feeding flexibility of the diverse groups included within COBALT's three zooplankton size classes (Fuchs and Franks, 2010; Hansen et al., 1994). The prey

availability is summarized in Table A2. In accordance with Van Oostende et al. (2018), grazing of the largest phytoplankton size class was limited primarily to the largest zooplankton (i.e., large-bodied copepods and krill), with only weak controls by medium-sized zooplankton. More flexibility was allotted across the smaller size classes.

To better represent light limitation in nearshore waters and/or waters under strong riverine influence, we enhanced the light attenuation in waters with salinity < 30 PSU or depth < 30 m to levels consistent with case 2 waters (Jerlov, 1976). This was

achieved by augmenting the background light attenuation in the Manizza (2005) scheme by 0.05 m$^{-1}$. This yields blue/green diffuse attenuation rates of about 0.1–0.3 m$^{-1}$ (transmittance of 90% to 75% per meter) for typical coastal chlorophyll levels between 0.5 and 5 mg chl m$^{-3}$, respectively. This augmentation to the light attenuation is currently only active for photosynthetic calculations in the biogeochemical model, and is acknowledged to be a simple first step toward more comprehensive representations of complex coastal ocean optics (e.g., Skákala et al., 2020). Its inclusion reflects the potential importance of

the depth of the euphotic zone for hypoxia in the Gulf of Mexico (Schaeffer et al., 2011).





To allow COBALT to better integrate riverine nutrient inputs that have nitrogen to phosphorus ratios (N:P) that often strongly depart from characteristic oceanic (i.e., Redfield) values, we augmented the phytoplankton N:P ratios for small, medium and large phytoplankton to include the potential for reduced P usage in low P environments (i.e., "P frugality"). This was parameterized with the emergent negative relationship between phytoplankton N:P and P concentration identified by Galbraith and Martiny (2015), enabling phytoplankton to achieve a maximum N:P ratio of 31 (nearly twice the canonical Redfield ratio) in low P environments. Minimum N:P ratios were restricted to characteristic values for each size class (Finkel et al., 2010) because the highest P regions in the NWA12 are generally associated with high N:P riverine inputs, which observations (e.g., Sterner and Elser, 2003; Hall et al., 2005) suggest would prevent the lower N:P ratios predicted by the Galbraith and Martiny (2015) relationship under phosphate rich conditions. The net effect of this simple phytoplankton N:P parameterization is thus to allow phytoplankton to better utilize excess N under low phosphate conditions, while reverting to characteristic N:P ratios under moderate and high phosphate levels. The addition of the fourth phytoplankton group and dynamic phytoplankton N:P ratios increases the number of prognostic state variables in COBALT from 33 in its standard formulation, to 40.

### 2.2.1 Biogeochemical model forcing

Open boundary data for biogeochemical tracers were set using several sources. Boundary data for $NO_3$, $O_2$, $PO_4$, and $SiO_4$ were obtained from climatologies in the World Ocean Atlas (WOA) 2018 dataset (Boyer et al., 2019). In the upper 800 m, we used seasonal climatologies from the WOA as boundary conditions, while below 800 m (where only long-term mean values are available) we used the long-term mean data. Boundary data for alkalinity (ALK) and dissolved inorganic carbon (DIC) were estimated from GLORYS monthly average temperature and salinity by applying the methods developed by Carter et al. (2021). We used the multiple linear regression version of the algorithm from Carter et al. (2021) that predicts ALK and DIC from potential temperature and salinity using a multiple linear regression, with an adjustment for the year. Carter et al. (2021) fit parameters for the regression over a 5° x 5° x 33 depth grid, and these coefficients were then interpolated to the NWA12 model boundaries and used to predict time-varying ALK and DIC from the interpolated GLORYS12 temperature and salinity. Boundary data for the remaining tracers, which generally have shorter turnover times, were set to 1993–2014 averages from the global COBALT simulation of Stock et al. (2014).

Biogeochemical tracers used the same tracer reservoir boundary condition scheme as temperature and salinity. However, as the external biogeochemical fields are not as well constrained by data as temperature and salinity and biogeochemical simulations are sensitive to even small amounts of spurious mixing or circulation near the boundary, we increased the inflow length scale to 300 km. This succeeded in imposing the desired lower frequency biogeochemical trends at monthly and longer time scales, while approaching a no-gradient condition for dynamics operating on daily timescales within the euphotic zone, thus dampening spurious biogeochemical signals. The impacts from such signals are further dampened by the large distance between the domain boundaries and the coastal regions of primary interest.

The primary source of river nutrient, carbon, and alkalinity data was the River Chemistry for the United States Coast (RC4USCoast) dataset compiled by Gomez et al. (2022). This product relies mainly on river chemistry observations from the United States Geological Survey (USGS) sub-selected for inputs to coastal waters and processed into monthly climatolo-





gies and, where possible, time series. The simulations herein used the 1990–2022 climatology to constrain dissolved organic carbon (DIC), alkalinity (ALK), nitrate ($NO_3$), ammonia ($NH_4$), phosphate ($PO_4$), the dissolved and particulate organic phases of N and P, oxygen ($O_2$), and silicate ($SiO_4$, derived from data on silicon dioxide $SiO_2$). 50% of particulate phosphorus was assumed to be mobilized in estuaries, with the remainder buried (Froelich, 1988; Sutula et al., 2004). Dissolved organic nitrogen and phosphorus inputs were fractionated to labile (40%), semi-labile (30%) and semi-refractory (30%) pools to be consistent with the range of bioavailability in Wiegner et al. (2006).

For Canadian waters, the river forcing described in Lavoie et al. (2021) was enlisted. This data included DIC, ALK, and nitrate. Other nutrient inputs were estimated by applying the ratio between dissolved inorganic nitrogen and $PO_4$, dissolved and particulate N and P from the semi-empirical GlobalNEWS2 algorithm (Mayorga et al., 2010) to the Lavoie et al. (2021) nitrate estimates. GlobalNEWS2 estimates were enlisted directly to constrain river inputs in Mexico, Central and South America. Oxygen for Canadian and Mexican/Central American/South American rivers was specified to be in equilibrium with climatological ocean temperatures at the river mouths. Following de Baar and de Jong (2001), iron concentrations in all rivers were specified to be 70 nanomolar.

RC4USCoast and other nutrient concentrations were mapped onto GloFAS freshwater inputs (See Section 2.1) using a nearest neighbor algorithm with larger rivers superseding smaller ones. Nutrient loads vary with river flows and month, but more subtle variations/trends in nutrient concentrations over the 1993–2019 model simulation period (e.g., increasing alkalinity in the Mississippi-Atchafalaya River System; Gomez et al., 2021) are omitted for the baseline configuration presented herein.

For the atmosphere, time- and latitude-varying atmospheric $CO_2$ concentration was set using the monthly historical time series from Meinshausen et al. (2017). We extended the historical $CO_2$ time series, which ends in 2014, using the atmospheric $CO_2$ concentration projected under the SSP2-4.5 emissions scenario (Meinshausen et al., 2020). Wet and dry deposition of $NO_3$, $NH_4$, and lithogenic dust were specified using a 1993–2014 monthly climatology from the historical simulation of GFDL's ESM4.1 earth system model (Dunne et al., 2020). As in ESM4.1 (Stock et al., 2020), iron deposition was approximated by assuming that dust is composed of 3.5% iron and iron solubility varies inversely with the surface dust concentration following Baker and Croot (2010). Dry deposition of phosphorus was also approximated from the climatology of dry dust deposition by assuming that dust consists of 563 ppm phosphorus of which 22% is bioavailable. These values were obtained from the global ocean averages of Herbert et al. (2018).

### 2.3 Model spinup and hindcast simulation

The primary model simulation evaluated in this paper is a hindcast simulation that was run from years 1993 to 2019 using the configuration and forcing described in previous sections. This simulation was initialized from rest on January 1, 1993 using GLORYS12 temperature and salinity for that date as the initial conditions. For the biogeochemical tracers, we ran a 10-year spinup beforehand to allow the biogeochemistry to adjust from the coarse climatological initial conditions, at least near the surface. This spinup simulation also started from rest in 1993 using initial conditions from the GLORYS12 reanalysis and the same biogeochemical data sources used to create the boundary conditions. We ran this simulation for 10 years using the forcings described previously. Carbon dioxide forcing was applied differently during this spinup simulation, however, to obtain



a model state closer to equilibrium with 1993 conditions: atmospheric $CO_2$ was applied by repeating the 1993 annual cycle

from the Meinshausen et al. (2017) forcing dataset, and open boundary condition ALK and DIC were calculated using the Carter et al. (2021) algorithm with the time fixed to 1993. We then started the main hindcast simulation on January 1, 1993 using the biogeochemical tracer fields at the end of the spinup simulation as initial conditions for the main simulation.

### 2.4 Model evaluation

To evaluate the utility of the NWA12 model for marine resource applications, we focused on three aspects of the model

performance. First, we considered general, basin-wide indicators of the model simulation fidelity, such as climatologies of surface temperature and macronutrients, and drivers of large-scale variability, such as the mean and variability of the position of the Gulf Stream. Second, we evaluated a high priority subset of features that are essential to simulate accurately in order to reproduce regional ecosystem variability, such as the temperature and salinity of water entering the Gulf of Maine at depth through the Northeast Channel, and fine-scale ecosystem responses to climate variability, such as hypoxia on the Louisiana-

Texas Shelf. For these two sets of model-data comparisons, a summary of the metrics considered and the datasets used as references is provided in Table 2. Third, we tested the computational cost and scaling of the model to assess the feasibility of using the model to run the long hindcasts and multi-member forecasts and projections needed to inform living marine resource management and other applications to coastal ocean decision-making.

Most physical ocean metrics were evaluated using the GLORYS12 reanalysis, which was also used as the model open

boundary conditions and reliably simulates regional hydrography (Castillo-Trujillo et al., 2022), as a reference dataset. We also included results from one or more in-situ or remote sensing observational datasets as an additional comparison where available. For most model-data comparisons, we calculated four quantitative skill metrics: bias (mean difference between model and data), root mean square error (RMSE), Pearson correlation coefficient (corr., or $r$), and median absolute error (MedAE, expressed as $\mathrm{Median}\left(|\mathrm{model}_i - \mathrm{data}_i|\right)$, which is robust against outlying errors). For chlorophyll, we used the Spearman rank

correlation instead of the Pearson correlation due to the nonlinearity of the data. All of these metrics were calculated using the xskillscore Python module (DOI:10.5281/zenodo.5173153). For spatial comparisons where the observed product had a similar resolution as the NWA12 model grid (finer than 1/4°), the observations were bilinearly interpolated onto the model grid. One exception is chlorophyll-$a$, which has fine resolution in the observed product (4 km) and high spatial variability and thus was conservatively interpolated onto the NWA12 model grid. For spatial comparisons where the observed product had a resolution

of 1/4° or coarser, the model data was conservatively interpolated onto the observed product grid. All comparisons except for the tidal sea surface height evaluation used monthly mean model output or longer-period averages calculated from monthly or daily means, so no processing was applied to remove tides from the model output.



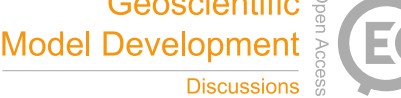

**Table 2.** Primary diagnostics used to evaluate the model, the time sampling of these diagnostics, and datasets used as references for evaluation of model performance.

| Metric | Time sampling | Reference datasets |
|---|---|---|
| Sea surface temperature | Annual and seasonal mean climatology | GLORYS12 |
| | | OISST v2 |
| | Annual mean climatology | GLORYS12 |
| Sea surface salinity | | NCEI Regional climatologies |
| Mixed layer depth | Winter (Jan–Mar) mean climatology | de Boyer Montégut (2004) |
| Gulf Stream position | | |
|     15 °C isotherm at 200 m | Annual mean climatology | GLORYS12 |
|     Mean latitude of maximum SSH variability | Annual mean climatology | Satellite altimetry |
|     SSH-based index of Gulf Stream position | Monthly time series | Satellite altimetry |
| | 25-month rolling average | |
| Mean sea level | Annual mean | GLORYS12 |
| | | Gridded satellite altimetry |
| M2 and K1 tidal amplitudes and phases | Estimated from hourly model data | TPXO9 |
| Surface nutrients | Seasonal mean climatology | World Ocean Atlas 2018 |
| Sea surface chlorophyll | Seasonal mean climatology | OC-CCI v6.0 |
| Mesozooplankton Biomass | Seasonal mean climatology | COPEPOD |
| Surface $pCO_2$ | Annual mean climatology | Bakker et al. (2016) |
| | Seasonal amplitude | Landschützer et al. (2020) |
| Surface ALK, DIC, $\Omega_{ar}$ | 2004–2018 mean | Jiang et al. (2022) |
| Sea surface temperature | Trend of annual mean 2005–2019 | GLORYS12 |
| | | OISST v2 |
| Bottom temperature in NE US EPUs | Annual average anomalies | NOAA Fisheries surveys |
| | | GLORYS12 |
| Deep water in Northeast Channel | Monthly mean time series | CTDs |
| | | Buoy N01 |
| | | GLORYS12 |
| Winter sea ice | Monthly climatology of concentration | Cavalieri et al. (1996) |
| | Time series of extent | |
| Mid-Atlantic Bight Cold Pool | June-September mean | du Pontavice et al. (2022) |
| Gulf of Mexico chlorophyll anomalies | Composite means of ENSO phases | OC-CCI |
| LA-TX shelf hypoxia | Seasonal climatology and July time series | Matli et al. (2020) |





### 2.4.1 Basin-wide indicators of model performance and drivers of variability

Model mean sea surface temperature (SST) was compared with the means of the 1/4° OISST v2 product derived from remote
sensing and in situ observations (Reynolds et al., 2007) and the GLORYS12 reanalysis. We compared the difference between
the 1993–2019 model mean with the same time period from the two reference datasets. We also evaluated model biases in the
seasonal SST climatology by comparing this metric to the OISST product.

We compared the 1993-2019 mean sea surface salinity (SSS) from the model with the corresponding mean SSS from the
GLORYS12 reanalysis and from the NOAA NCEI regional ocean climatologies, which are interpolated and quality-controlled
climatologies at 1/10° resolution derived from the World Ocean Database (Seidov et al., 2018, 2019). For the NCEI regional
climatologies, which are provided as decadal averages with the most recent decade extended to 2017, we used the weighted
average of the 1995–2004 and 2005–2017 means.

MOM6 computed the model mixed layer depth (MLD) online using the common definition of the mixed layer bottom as
the depth where the potential density difference relative to the surface first reaches 0.03 kg m$^{-3}$ (Griffies et al., 2016). We
compared the winter (January, February, March) mean MLD with the long-term MLD climatology derived from profiles in the
World Ocean Database and Argo datasets by de Boyer Montégut (2004). We used the November 2022 update to this dataset
(DOI:10.17882/91774). The MLD in this dataset is also defined as the depth where the density is 0.03 kg m$^{-3}$ greater than the
surface density; however, it uses 10 m depth as the surface, whereas MOM6 uses the surface layer (between 0–2 m here). This
will introduce a small bias to the comparison.

The mean and variability of the Gulf Stream position was calculated using two different metrics, one using temperature
and the other using sea surface height (SSH). First, we calculated the position of the north wall of the Gulf Stream based
on the position of the 15 °C isotherm at 200 m depth in the 1993–2019 mean temperature (Chi et al., 2019; Fuglister and
Voorhis, 1965). This metric was also calculated using the GLORYS12 reanalysis for comparison. Second, along meridional
lines between 72 °W and 52 °W spaced 1° longitude apart, we calculated the latitudes where the variance of monthly mean
sea surface height anomaly (the difference between the monthly mean SSH and the 1993–2019 calendar month mean) was
highest. These latitude/longitude points form the mean position of the Gulf Stream. An index representing north/south shifts
of the Gulf Stream over time was calculated by averaging the sea level anomaly at the mean Gulf Stream position points
and then dividing it by the standard deviation of the average sea level anomaly. Although short term eddy variability will
introduce some noise to this metric, longer-term means generally represent broad, coherent shifts in the Gulf Stream position,
and thus we also calculated a 25-month centered rolling average of the index. These metrics for SSH-based mean position and
variability are adapted from Pérez-Hernández and Joyce (2014) and are the same as those used in the NOAA Fisheries State
of the Ecosystem (SOE) Report for the Mid-Atlantic (NOAA Fisheries, 2022a). This metric was also calculated using monthly
mean absolute dynamic topography from the gridded satellite altimetry dataset provided by the Copernicus Marine Service
(DOI:10.48670/moi-00148). We also compared long-term mean sea surface height in the broader western boundary current
region using both the GLORYS12 and satellite altimetry datasets.





The model simulation of tidal elevations was assessed by calculating the amplitudes and phases of the largest semidiurnal (M2) and diurnal (K1) constituents for each model grid point from hourly model output. Due to the computational time and storage costs of saving hourly model output, this assessment was only done for one year of model simulation started in 1993 with the same initial conditions as the main hindcast. This simulation did not include the COBALT biogeochemical component

to reduce the computational cost and given the expected negligible feedback of biogeochemistry on tides. Modulation of the tides by the 18.6 year nodal cycle was also disabled for this run so that the modulation did not need to be re-estimated and corrected for when calculating tidal constituents properties from the model output. The amplitudes and phases of the M2 and K1 constituents were calculated at every grid point in the model using the UTide Python package (Codiga (2011); https://github.com/wesleybowman/UTide) and were compared with corresponding amplitudes and phases from the TPXO tide

data used as tidal boundary conditions in the NWA12 model.

Model predicted surface chlorophyll-*a* (chl-a) was compared with satellite remote sensing estimates from the Ocean Colour Climate Change Initiative (OC-CCI) dataset version 6.0 (Sathyendranath et al., 2019), which merges estimates of ocean surface chl-a from multiple satellites and algorithms into one consistent product. We compared the model and OC-CCI seasonal climatologies calculated over 1998–2019.

The model seasonal mean climatology of mesozooplankton biomass integrated over 0–200 m was compared with observations from the COPEPOD dataset (Moriarty and O'Brien, 2013). Mesozooplankton in the COBALT model consists of the medium (200-2000 $\mu$m Equivalent Spherical Diameter) and large (2000-20000 $\mu$m ESD) size classes. The COPEPOD dataset reports biomass adjusted to be consistent with measurements from a 333 $\mu$m mesh (Moriarty and O'Brien, 2013), which is likely to exclude a significant fraction of sizes on both the small and large end of COBALT's zooplankton size range (Skjoldal

et al., 2013). Shropshire et al. (2020), for example, found that mesozooplankton biomass in 333 $\mu$m mesh nets in the Gulf of Mexico was approximately half (0.5093) that measured in 202 $\mu$m mesh nets. This is similar to the 0.6195 adjustment found by O'Brien (2005), and Skjoldal et al. (2013) further suggest that an even smaller 150 $\mu$m mesh net would be better in coastal systems. Given these uncertainties, we multiplied COPEPOD biomass estimates by 2 before comparing with COBALT. We recognize the possibility that escape by larger zooplankton size classes and inefficient net sampling of gelatinous zooplankton

likely make the adjusted observations a lower bound for mesozooplankton biomass across the size range covered by COBALT.

Model seasonal surface nutrient climatologies were assessed by comparing modeled surface $NO_3$ and $PO_4$ with climatologies from the World Ocean Atlas (WOA) 2018 (Boyer et al., 2019).

Model surface $pCO_2$ was compared with observations derived from the Surface Ocean CO2 Atlas database version 2021 (SOCATv2021, Bakker et al. (2016)) during 1993–2019 and against a $pCO_2$ data-product generated from SOCAT observations

by a two-step neural network interpolation (data-product, Landschützer et al. (2020)) during 1998–2015. We evaluated the model and observed seasonal variability of $pCO_2$ which is expressed as the root mean square of the monthly $pCO_2$ anomalies (RMS, in $\mu$atm). Due to the discontinuous $pCO_2$ observations in time in the SOCAT database, the evaluation of the model in reproducing the seasonal $pCO_2$ variability was only performed against the data-product.

Model surface total alkalinity, DIC, and aragonite saturation state were compared with long-term observed means from the

dataset of Jiang et al. (2022). This dataset applies an objective analysis technique to observations from the Coastal Ocean





Data Analysis Product in North America (CODAP-NA; Jiang et al., 2021) and Global Ocean Data Analysis Product version 2 (GLODAP v2; Lauvset et al., 2021) to produce a 1° gridded long-term mean for each variable. The model means were calculated over years 2004–2018 to match the time period of the observations from CODAP-NA.

### 2.4.2 Relevant regional features and ecosystem responses

Since the early 2000s, the ocean along the Northeast United States has warmed faster than the vast majority of all other ocean regions (Pershing et al., 2015; Seidov et al., 2021). We assessed the ability of the model to reproduce this warming and other temperature trends throughout the model domain by calculating the 2005–2019 annual mean sea surface temperature trend at each point in the model and comparing with trends from the GLORYS12 and OISST datasets used in Section 2.4.1. 2005 was chosen as the start date of the trend analysis to match the date chosen by Seidov et al. (2021), and is one year later than the
date chosen by Pershing et al. (2015).

Next, we assessed three metrics related to bottom temperature on the Northeast U.S. Continental Shelf that are used to inform management in the NOAA Fisheries State of the Ecosystem (SOE) Reports for New England (NOAA Fisheries, 2022b) and the Mid-Atlantic (NOAA Fisheries, 2022a): (1) bottom temperatures averaged within four ecological production units (EPUs), (2) deep temperature and salinity with the Gulf of Maine Northeast Channel and associated water masses, and (3) bottom
temperature within the Mid-Atlantic Bight cold pool region. Data for the observed metrics were obtained directly from the SOE reports, and metrics from the model data were calculated using nearly identical methods.

First, New England and Mid-Atlantic fisheries management EPUs are defined for the southwestern Scotian Shelf, Gulf of Maine, Georges Bank, and the Mid-Atlantic Bight (Fig. 1b). Within each EPU, area-weighted model average bottom temperatures were computed for all model tracer points falling within each EPU, and anomalies were calculated by subtracting the
1993–2010 monthly climatology for each EPU. For comparison, EPU-average bottom temperature anomalies were extracted from the GLORYS12 reanalysis using identical methods. The model bottom temperature anomalies were also compared with observations obtained directly from data used in the SOE reports. These observed anomalies were calculated using slightly different methods. Observed bottom temperatures were derived from CTD profiles collected during routine surveys that were performed at least twice per year (most often in the late spring and autumn months). The annual harmonic method described
in Mountain (1991), in which the climatology is a simple sine wave with a period of one year, was used to calculate these anomalies because it works well at extracting anomalies from this sparse and irregular data. Finally, for all three datasets, we calculated annual averages of the monthly anomalies.

Second, we assessed the model's ability to simulate water masses entering the narrow, deep Northeast Channel in the Gulf of Maine that drive ecosystem-relevant temperature and salinity variability within the Gulf. Using methods from Mountain
(2012) and the NOAA Fisheries State of the Ecosystem Reports for New England (NOAA Fisheries, 2022b), we evaluated the model monthly mean potential temperature and salinity averaged in the Channel between 42.2–42.6 °N, 66.0–66.8 °W and 150–200 m depth. Temperature and salinity in the channel is influenced by mixing between relatively cool, fresh Scotian Shelf Water (defined by Mountain (2012) as T=2 °C, S=32), moderately warm, salty Labrador Slope Water (T=6 °C, S=34.6), and very warm, salty Warm Slope Water (T=12 °C, S=35.4). Recent observations have shown a significant increase in Warm Slope



Water and a corresponding decrease in Labrador Slope Water (Balch et al., 2022; NOAA Fisheries, 2022b), so we used annual
mean potential temperature and salinity to calculate time series of the composition of the water in the channel in percentages
of these two masses using methods from NOAA Fisheries (2022b). These metrics calculated from the model were compared
with matching metrics determined using potential temperature and salinity from three different products: the database of CTD
profiles used in NOAA Fisheries (2022b), the GLORYS12 reanalysis, and the N01 buoy from the Gulf of Maine Moored Buoy
Program (Wallinga et al., 2003) (available at 180 m depth from June 2004–July 2017).

Third, the ability of the model to simulate the cool bottom water along the shelf that forms the Mid-Atlantic Bight cold
pool was assessed by comparing June–September mean bottom temperatures in the model and GLORYS12 reanalysis and
calculating the cold pool index developed by du Pontavice et al. (2022) and reported in NOAA Fisheries (2022a). This index
uses the June-September bottom temperature anomaly averaged over a common cold pool region defined by depth, average
temperature, and latitude and longitude; see du Pontavice et al. (2022) for additional details. The index is higher when the
bottom is warmer than average and the cold pool is smaller than average. du Pontavice et al. (2022) calculated the index using
the GLORYS12 reanalysis from 1993–2019 and extended the index farther back in time using a bias-corrected ocean model
simulation. For historical context, we show the full time series; however, the du Pontavice et al. (2022) index is only derived
from GLORYS12 during the time period when the model and data overlap.

The coastal ocean in the far northern portion of the model domain, including the Gulf of St. Lawrence and the coast of
Labrador and Newfoundland, partially freezes over in the winter. We assessed the ability of the coupled SIS2 sea ice model
to simulate the spatial and temporal variability of sea ice cover in this region by comparing the model output with satellite
observations of sea ice concentration from the National Snow and Ice Data Center (NSIDC) (dataset NSIDC-0051; Cavalieri
et al. (1996)). First, we compared the model and satellite monthly climatologies of sea ice concentration over 1993–2019.
Second, we compared time series of the areal extent of sea ice in the Gulf of St. Lawrence by determining the total area with a
monthly sea ice concentration above 15% for each month in the model and satellite data.

Moving the regional focus to the southern portion of the domain, the El Niño-Southern Oscillation (ENSO) has been shown
to drive marine resource-relevant variability of phytoplankton biomass in the Gulf of Mexico. Winters and springs when an
El Niño occurs are associated with increased river discharge, stronger mixing, and changes in coastal currents, which tend to
produce increased phytoplankton and surface chlorophyll in the northern Gulf of Mexico, while La Niña winters and springs
show an opposite but weaker response (Gomez et al., 2019). We conducted an analysis similar to Gomez et al. (2019) to
determine if the model could reproduce this mode of variability. We compared composite means of model surface chlorophyll
during winter (December–February) and the following spring (March–May) in El Niño and La Niña years with composites
calculated from the OC-CCI remote sensing estimates for the same years. El Niño events were defined as those where the
3-month running average SST anomaly in the Niño-3.4 region, published as the ONI index by the NOAA Climate Prediction
Center, exceeded 0.5 °C in December and remained positive in the following spring, and La Niña years were defined as those
where the ONI index was below -0.5 °C in December and remained negative in the following spring. During the period where
both the model and OC-CCI data are available (winter 1997–2019), six El Niño events occurred (1998, 2005, 2010, 2015,





2016, 2019) and nine La Niña events occurred (1999, 2000, 2001, 2006, 2008, 2009, 2011, 2012, 2018); note that the year of
an event is defined here as the year following the occurrence of the December ENSO anomaly.

Finally, the area along the Louisiana-Texas shelf with bottom hypoxia (bottom dissolved oxygen concentration below 2 mg
$l^{-1}$) was calculated from model daily mean bottom oxygen and compared with data from the geostatistical model of Matli et al.
(2020) that combines observations, ocean model, and atmospheric data into a reliable estimate of hypoxic area. We integrated
the model hypoxic area over the same region as Matli et al. (2020) (between 89.512-94.605 °W and 28.219–29.717 °N with a
depth of 100 m or less). Model performance was assessed by comparing monthly climatologies of hypoxic area averaged over
the years where the model and geostatistical estimates overlap (1993–2017) and by comparing the time series of monthly mean
hypoxic volume for July (the normal peak month of hypoxic area).

### 2.4.3 Computational implementation and scaling

To determine the feasibility of using the model to provide future predictions and projections with information about uncertainty
to support living marine resource applications, we assessed the total run time of the model relative to approximate values needed
to support application needs and relative to the number of processing elements (PEs) used. The primary configuration of the
model evaluated in this paper was run using a 50x50 layout, which divides the 775x845 model grid across a 50x50 grid of
PEs, yielding a 16x17 chunk of the model domain on a typical PE. The MOM6 land masking feature, which eliminates PEs
that do not contain any ocean points, was enabled, resulting in only 1646 PEs actually used (a savings of 34%). To assess the
scalability of the model across other layouts, we determined the time needed to run 1 year of simulation using layouts of 40x40,
50x50, 60x60, and 70x70 PEs (because the model domain is nearly square in terms of grid points, we only considered square
decompositions). We focused on the total run time of the model, including the time needed for initialization, running the main
loop, and writing output, since this time ultimately determines the computational tractability of the model. Each 1 year timing
simulation was repeated three times and the run times were averaged to obtain a more accurate result. In addition to assessing
the time needed to run one year of simulation, we evaluated the model efficiency by dividing the total computational cost of
the run (the number of PEs times the time spent running the simulation) using the 40x40 layout by the computational cost of
each of the other layout experiments. The ideal efficiency of 1 would indicate that the run time scales exactly with the inverse
of the number of PEs (i.e., a configuration with twice the number of PEs would run in half the time), while efficiencies below
1 indicate that the model consumes more resources to complete a given simulation as the model domain is distributed across
greater numbers of PEs (even though it may still complete the simulation in less wall clock time).

We also assessed the benefit of a key feature of MOM6: the ability to separate the time steps for the ocean barotropic and
baroclinic dynamics from the time step for tracer advection, thermodynamics, mixing, and coupled ocean biogeochemistry. The
"thermodynamics" time step for the latter set of processes can be run with a significantly longer time interval than the dynamics
for computational speed. The basic model configuration presented in this paper uses an 1800 second thermodynamics time step
and a 600 second baroclinic dynamics time step. We assessed the benefits of this configuration in terms of computational cost
by repeating the 60x60 and 70x70 layout experiments discussed previously using a 600 s time step for both thermodynamics
and baroclinic dynamics, with all other configuration options identical (note that the 40x40 and 50x50 configurations cannot





complete a year of simulation with this shorter time step before the reaching the wall clock time limit per job imposed by our high-performance computing system).

These timing simulations were run on NOAA's "Gaea" high performance computing system using the c4 partition that has 36 Intel Broadwell processor cores per node. All model source code was compiled using "production" mode, which enables aggressive compiler optimizations. The model source code is archived at https://doi.org/10.5281/zenodo.7893349.

## 3   Results

### 3.1   Basin-wide indicators of model performance and drivers of variability

The MOM6-COBALT-NWA12 model accurately simulates the broad patterns of mean sea surface temperature throughout the domain, with moderate biases that are most prominent in the vicinity of the Gulf Stream (Figs. 2–3). These biases include a region of warm SST concentrated along the shelf break from Cape Hatteras to the Grand Banks and a region of cool SST to the south. The strongest cool bias in the model is found east of the Grand Banks where the North Atlantic Current takes a northward turn, which is consistent with the effects of an underestimation of the eastward extension of the Gulf Stream and the northward turn of the current that is often seen in ocean models across a wide range of spatial resolutions (e.g., Bryan et al., 2007; Scaife et al., 2011; Sein et al., 2017). The cool bias in MOM6-NWA12 is substantially less than the biases of over -5° found in 1/4° and 1/10° global climate models using a previous version of MOM, although some of the warm biases near the coast are greater (Saba et al., 2016). Outside of the western boundary current region, the model SST bias is generally minor. However, the SST biases are consistently negative between $-0.5$ and 0 °C, and the overall area-weighted mean bias is $-0.23$ °C compared to OISST. The spatial pattern of the model SST bias is consistent across seasons (Fig. 3); however, the magnitude of the bias is higher in winter and spring and lower in summer and autumn.

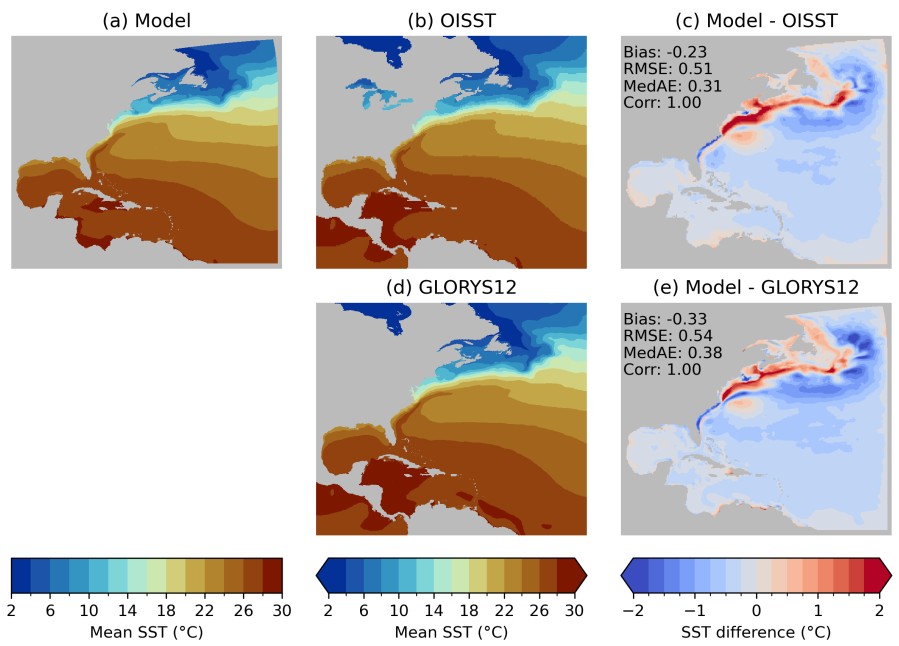

**Figure 2.** 1993–2019 mean sea surface temperature in the model (a) compared with the OISST (b) and GLORYS12 (d) datasets, and the difference between the model and the datasets (c, e).





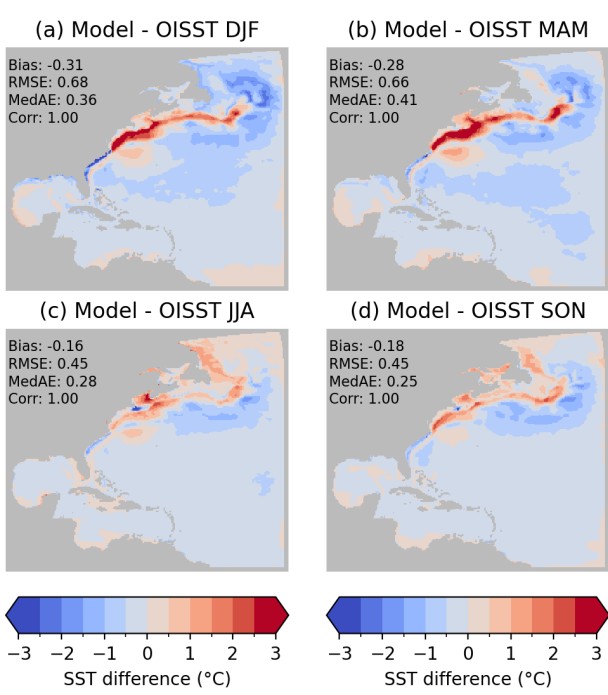

**Figure 3.** Difference between the 1993–2019 seasonal mean sea surface temperature in the model and the OISST dataset.

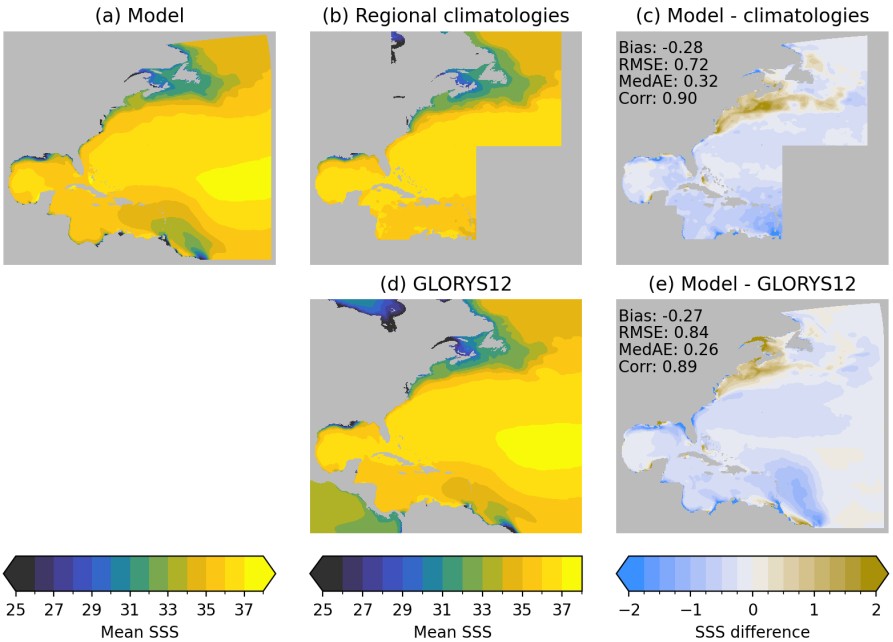

**Figure 4.** 1993–2019 mean sea surface salinity in the model (a) compared with the regional climatologies (b) and GLORYS12 (d) datasets, and the difference between the model and the datasets (c, e).

In the patterns of model mean sea surface salinity and bias (Fig. 4), a positive surface salinity bias is present along the shelf from Cape Hatteras to the Gulf of St. Lawrence, in the same region as the positive SST bias. The co-occurrence of these biases resembles, but is much milder than, the biases often seen in models with a poorly resolved Gulf Stream (e.g., Saba et al., 2016).
Elsewhere in the domain, the patterns in the model bias suggest some errors in the volume and placement of rivers or in the river plumes and coastal currents that carry riverine freshwater. For example, model salinity is biased low to the west of the Mississippi River and high to the east. It is worth noting that at 1/12°, the model will not resolve the first Rossby radius of deformation along the coastal ocean where the rivers discharge (Hallberg, 2013; Piecuch et al., 2018), and this will limit the ability to capture the dynamics of the river plumes. Overall, the area-weighted model mean salinity is biased low by 0.27–0.28
units compared to the observational and reanalysis datasets.

The model reproduces the broad patterns of winter mixed layer depths when compared to estimates derived from profiles (Fig. 5). The spatial correlation coefficient for the mixed layer depth based on a density difference from the surface of 0.03 kg m$^{-3}$ is high (0.94) and the mean bias is negligible (0.90 m). The RMSE of 22.37 m is somewhat high relative to the typical value; however, the high correlation and low median absolute error of 8.02 m suggests the higher RMSE may be primarily due
to mismatches in deep mixed layers. The model also has a tendency to predict winter mixed layers that are too deep to the north of the Gulf Stream along the shelf break, consistent with the temperature and salinity biases shown previously.





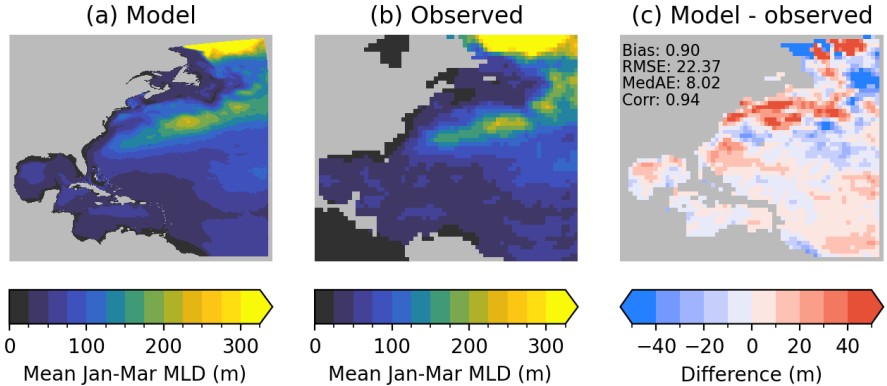

**Figure 5.** Winter (January-March) mixed layer depth in the model (a) and the observation-based climatology of de Boyer Montégut (2004), and the difference between the two (c).

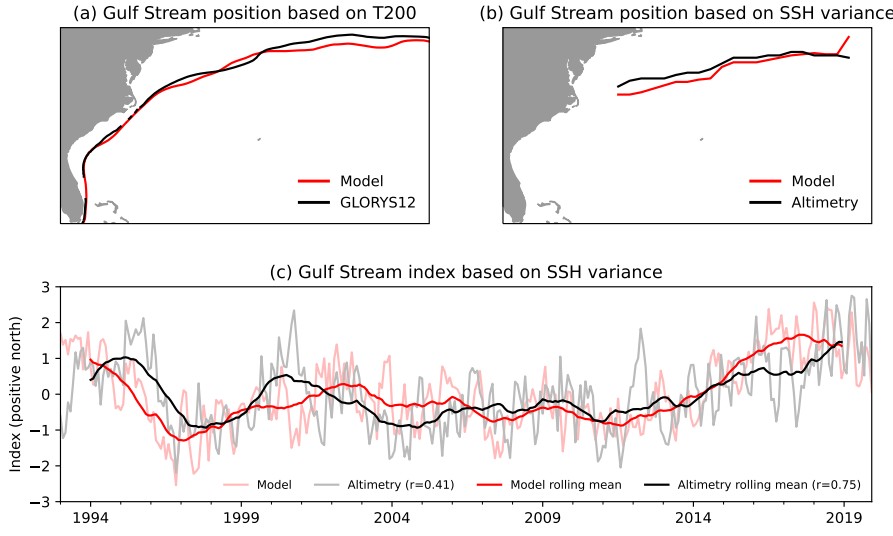

**Figure 6.** Evaluation of model Gulf Stream position based on the 15 °C isotherm at 200 m (a), the latitude of maximum sea surface height variance (b), and an index based on anomalies in the latitude of maximum SSH variance (c).

Although the previous metrics suggest the presence of a northward bias to the typical Gulf Stream path, which would bring excessively warm and salty water to the Northeast U.S. shelf, both of the metrics for the position of the Gulf Stream show that the average position is remarkably close to observations (Fig. 6). The 15 °C isotherm of mean temperature at 200 m displays the canonical separation from the shelf at Cape Hatteras in both the model and GLORYS12 reanalysis (Fig. 6a). This metric actually suggests the presence of a slight southward bias to the model Gulf Stream path, particularly downstream of the separation, which is also seen in the metric based on SSH variance (Fig. 6b).





At low frequencies, north-south shifts in the position of the Gulf Stream are well-simulated by the model, with a correlation of 0.75 for the 25-month rolling mean Gulf Stream index (Fig. 6c). This simulation is particularly good considering that shifts in the Gulf Stream path are typically difficult to capture even in data-assimilative reanalysis products (Chi et al., 2018). The model accurately places the Gulf Stream farther south than usual (indicated by a negative index) in the late 1990s with anomalously northern positions in the early 1990s and 2000s. The modeled and observed position was relatively constant during 2006–2013. Finally, the model roughly reproduces the northward shift in the Gulf Stream that occurred starting around 2014, although the shift occurs faster in the model. At higher frequencies, fluctuations in the model simulated position only loosely track the observed fluctuations associated with individual eddies and meanders—the correlation between the model and observations decreases from 0.75 for the 25-month rolling averages to 0.41 for the monthly values. However, this is not unexpected because the model boundaries are far from the region of interest and the model does not assimilate observations (i.e., eddies and meanders are present, but the formation and evolution of individual observed eddies are not deterministically simulated).

The spatial pattern of mean sea surface height in the northwest part of the model domain is consistent with Gulf Stream biases and the good performance in other parts of the domain (Fig. 7). Mean SSH closely resembles the reanalysis and satellite altimetry datasets from the Loop Current through the Florida Strait and up to Cape Hatteras. Beyond Cape Hatteras, the SSH pattern in the model has a weaker cross-shore gradient with more variability, the northwestern recirculation gyre (normally located offshore of the southern Mid-Atlantic Bight and visible as a region of low SSH) appears to be absent, and the Gulf Stream does not extend as strongly to the east and weakly curves to the north along the Grand Banks. These biases are consistent with the previous results: although the model simulates the mean and low frequency variability of the Gulf Stream path near the separation point well, the weaker and more variable extension, a weaker northwestern turn along the Grand Banks, and absent recirculation gyre combine to produce temperature and salinity biases with opposing patterns north and south of the mean Gulf Stream.

Semidiurnal and diurnal tides are well simulated in the model, with amplitude RMSEs that are low compared to the typical amplitude (RMSE of 5.44 cm for the semidiurnal M2 amplitude and 1.29 cm for the diurnal K1 amplitude) and high spatial correlations (0.95–0.96) (Figs. 8–9). Amplitude errors for the M2 tide are highest in the Gulf of Maine and Gulf of St. Lawrence (Fig. 8). In the Gulf of Maine, the model M2 amplitude is too low in the Bay of Fundy and too high in the southwest portion of the Gulf. M2 amplitudes are too high throughout the coastal Gulf of St. Lawrence, although the central amphidrome (tidal node) is well placed in the model. The contour lines of the model M2 phase in the Caribbean Sea show a patchwork pattern. A closer look revealed a pattern of tidal amplitude similar to the surface expression of internal tides generated along the Windward Islands and in the passage between Puerto Rico and the Dominican Republic found by Zaron (2019) (not shown). For the diurnal K1 tide, the amplitude is also too high in the Gulf of St. Lawrence and along the majority of the U.S. East Coast and in the Western Gulf of Mexico (Fig. 9). The phase of the model K1 tide is shifted by about an hour, particularly in the interior of the domain away from the boundaries.

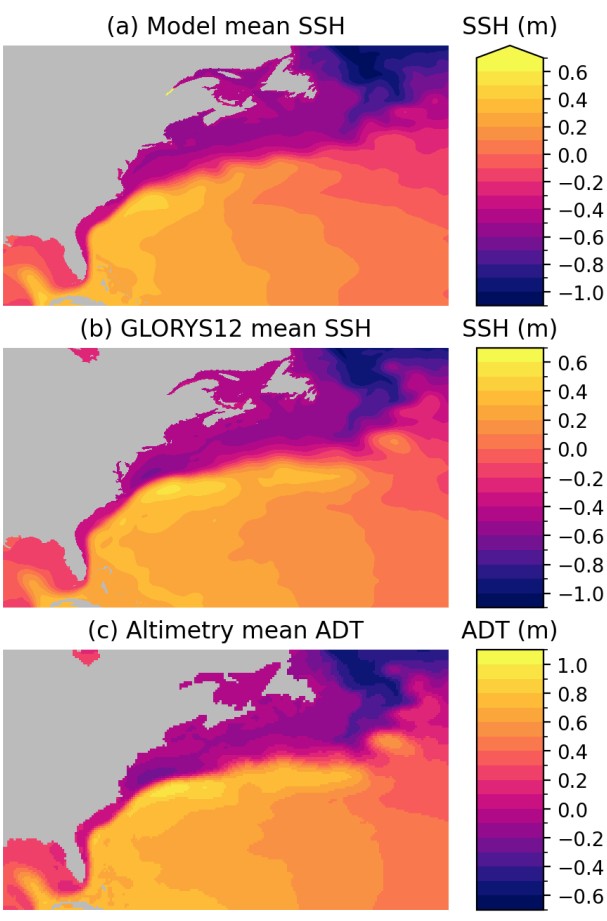

**Figure 7.** 1993-2019 mean sea surface height in the model (a) and the GLORYS12 reanalysis (b) and mean absolute dynamic topography from the gridded satellite altimetry dataset (c). Note that the color scale for (c) is shifted by 0.4 m.



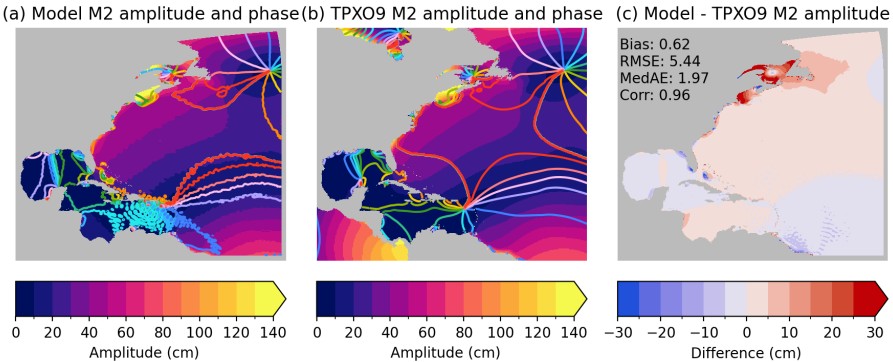

**Figure 8.** Semidiurnal M2 tidal amplitude and phase estimated from hourly sea level output from the model (a) compared with the reference TPXO9 tidal model data (b) and the difference between the model and reference M2 amplitude (c). Phase in panels a-b is indicated by a different colored line for each hour.

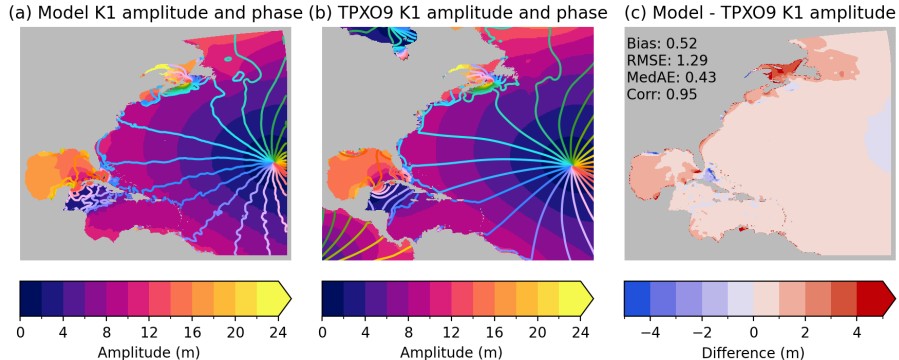

**Figure 9.** Diurnal K1 tidal amplitude and phase estimated from hourly sea level output from the model (a) compared with the reference TPXO9 tidal model data (b) and the difference between the model and reference K1 amplitude (c). Phase in panels a-b is indicated by a different colored line for each hour.

Shifting to biogeochemical patterns, the simulated surface nitrate (Fig. 10) and phosphate (Fig. 11) exhibit a robust large-scale nutrient drawdown from winter into the summer months. Winter nitrate, however, has a moderate high bias in shelf-adjacent waters in the northeastern part of the domain aligned approximately with high MLD biases in this region (Fig. 5). This winter high bias delays the depletion of nitrate in the spring. The model and observations largely converge in the summer before

615 high biases tend to re-emerge in the autumn with the onset of deeper mixing. Phosphate shows a similar winter/spring high bias as nitrate, as would be expected from a bias rooted in overly deep mixed layers along the shelf break. In contrast with nitrate, both the modeled and observed phosphate remain elevated in summer months in the coastal waters of the northeast United States and Canada. Modeled phosphate, however, is somewhat lower than that observed. In the Gulf of Mexico, simulated summer phosphate surpluses are not as high as those that WOA suggests. The proximity of observed highs with some of the

620 larger river systems in the southern Gulf of Mexico (e.g., the Panuco, Usamacinta, Coco) and Central and South America (e.g., Magdalena and Orinoco) suggest that part of this misfit may be attributable to uncertain river inputs in this region. In addition to the limited availability of river input data, ocean biogeochemical observations in the southern Gulf of Mexico are also sparse (Estrada-Allis et al., 2020), which introduces additional uncertainty to the comparison.

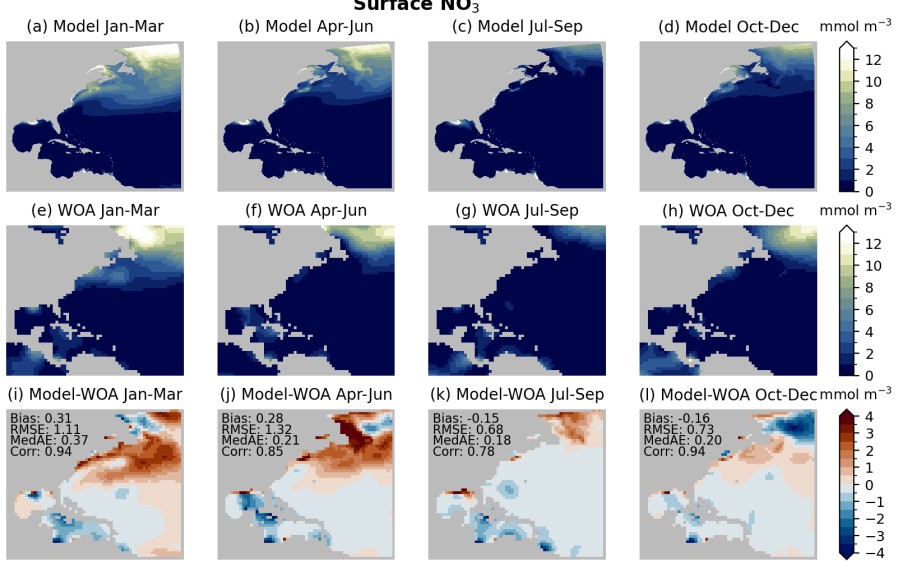

**Figure 10.** Comparison of model seasonal mean surface nitrate (a, d, g, j) with the World Ocean Atlas (b, e, h, k), and the difference between the datasets (c, f, i, l).





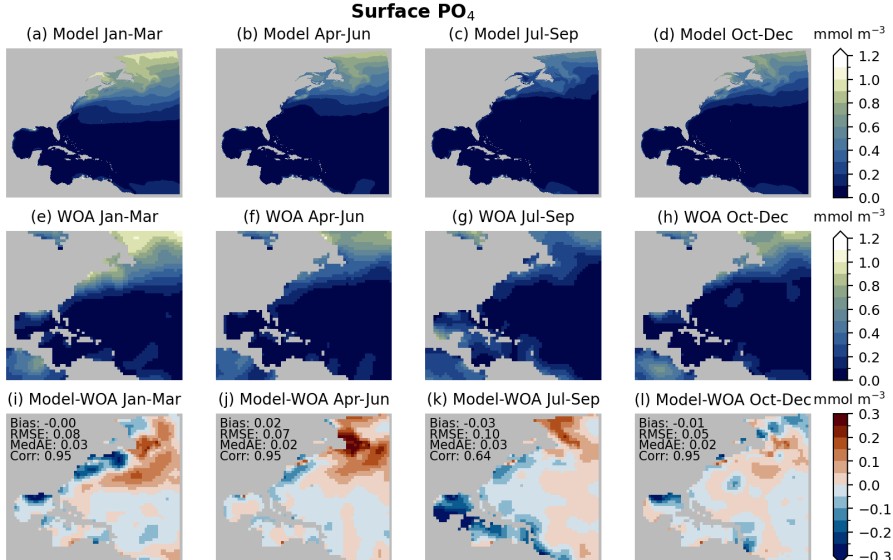

**Figure 11.** Comparison of model seasonal mean surface phosphate (a, d, g, j) with the World Ocean Atlas (b, e, h, k), and the difference between the datasets (c, f, i, l).

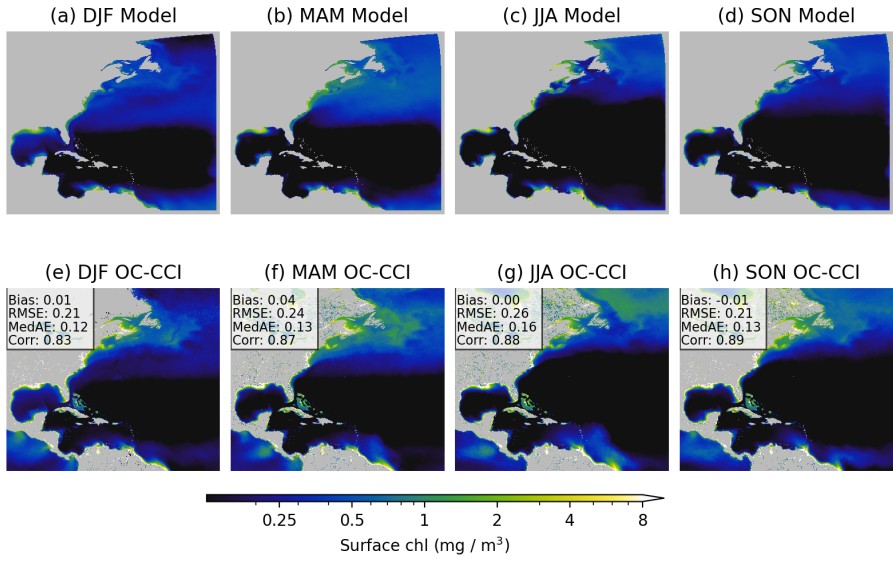

**Figure 12.** Evaluation of model seasonal mean surface chl-a (a–d) compared with OC-CCI satellite remote sensing estimates (e–h). Skill metrics were calculated for $\log_{10}$ transformed chlorophyll.

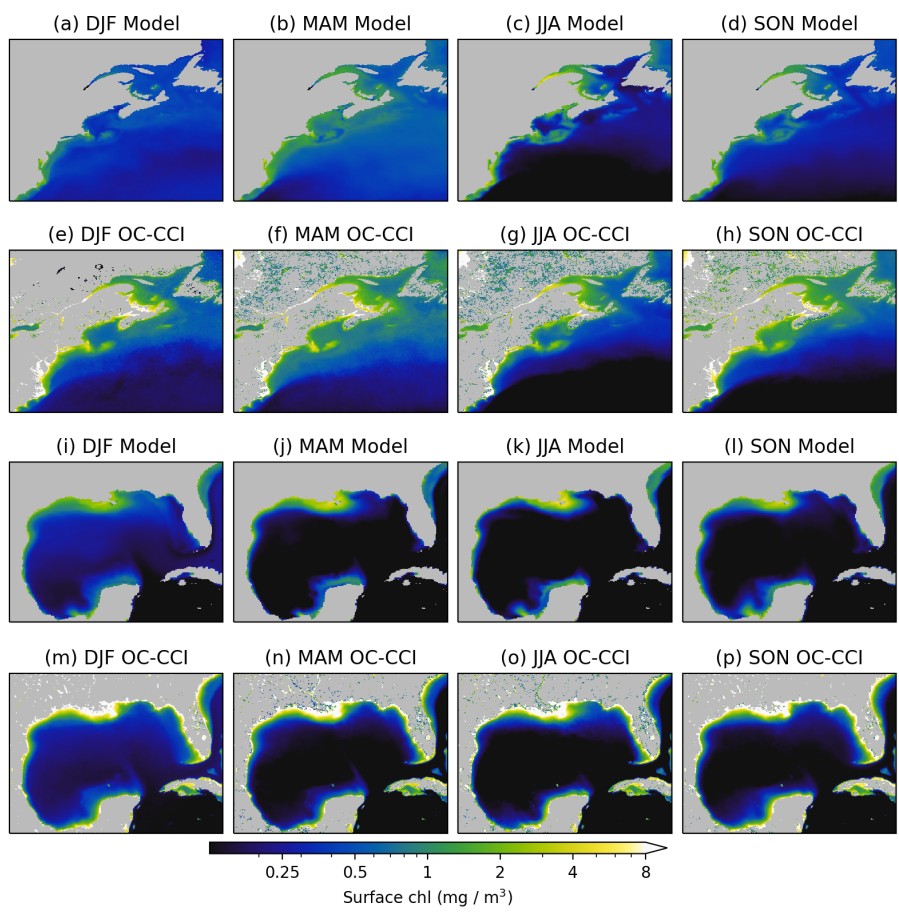

**Figure 13.** Evaluation of model seasonal mean surface chl-a (a–d, i–l) compared with OC-CCI satellite remote sensing estimates (e–h, m–p) over the Northeast U.S. and Gulf of Mexico.





The mean seasonal patterns of surface chl-a are simulated well by the model, despite some regional biases (Figs. 12–13). Spatial correlations (of $\log_{10}$ transformed chlorophyll) between the model and OC-CCI remote sensing estimates range from 0.83 in winter to 0.89 in autumn, and RMSEs in $\log_{10}$ units range from 0.21 in winter to 0.26 in summer, corresponding to a factor of 1.6–2.0 error. The chlorophyll minimum within the oligotrophic subtropical gyre is consistent with the satellite estimates, although chlorophyll is slightly higher than observed at the northern and southern limits of the gyre. Along the coast, in the Northeast U.S. and Gulf of St. Lawrence (Fig. 13a–h), seasonal patterns and cross-shore gradients are reproduced reasonably well, including the consistent chlorophyll hotspot on Georges Bank associated with sustained tidal mixing (Franks and Chen, 1996; Hu et al., 2008), the summer–autumn pattern of lower chlorophyll in the central Gulf of Maine ringed by higher chlorophyll along the coasts, high chlorophyll concentration in the Lower St. Lawrence Estuary and southern Gulf of St. Lawrence and a chlorophyll maximum that shifts from the spring in the Gulf of St. Lawrence to summer in the Estuary (Laliberté and Larouche, 2023). Spatial and seasonal patterns of surface chlorophyll are also simulated well in the Gulf of Mexico (Fig. 13i–p). Although seasonal and spatial patterns are reproduced well, in many of the coastal regions, including the northern Gulf Coast and the Northeast U.S and Gulf of St. Lawrence, modeled surface chlorophyll concentrations are substantially lower than the satellite estimates. However, we note that satellite based chlorophyll estimates are less reliable and more difficult to interpret in these turbid nearshore environments (Schofield et al., 2004; Dierssen, 2010), and satellite estimates in the northern regions of the model domain, including the Gulf of St. Lawrence, are also affected by sea ice and low sun angles in the winter that limit the availability of data (Laliberté and Larouche, 2023).

Looking up the foodweb, the simulated seasonal mean mesozooplankton biomass (Fig. 14) also shows seasonal and spatial gradients consistent with net data, though values in the South Atlantic Bight are higher than the relatively sparse observations in the region suggest. On average, the model is biased high even after adjusting for a factor of 2 under-sampling of the full mesozooplankton community. However, as discussed in the methods, this correction still neglects escapement by large mesozooplankton and potential contributions by undersampled gelatinous components. In the offshore waters of the subtropical gyre, mesozooplankton biomass generally falls to around 1–3 mg C m$^{-3}$ (200–600 mg C m$^{-2}$). This is considerably larger than the 330 micron mesh net estimates included in the COPEPOD database, but more comparable to smaller mesh sampling at the Bermuda Atlantic and Hawaii Ocean Time Series (e.g., Roman et al., 2001).



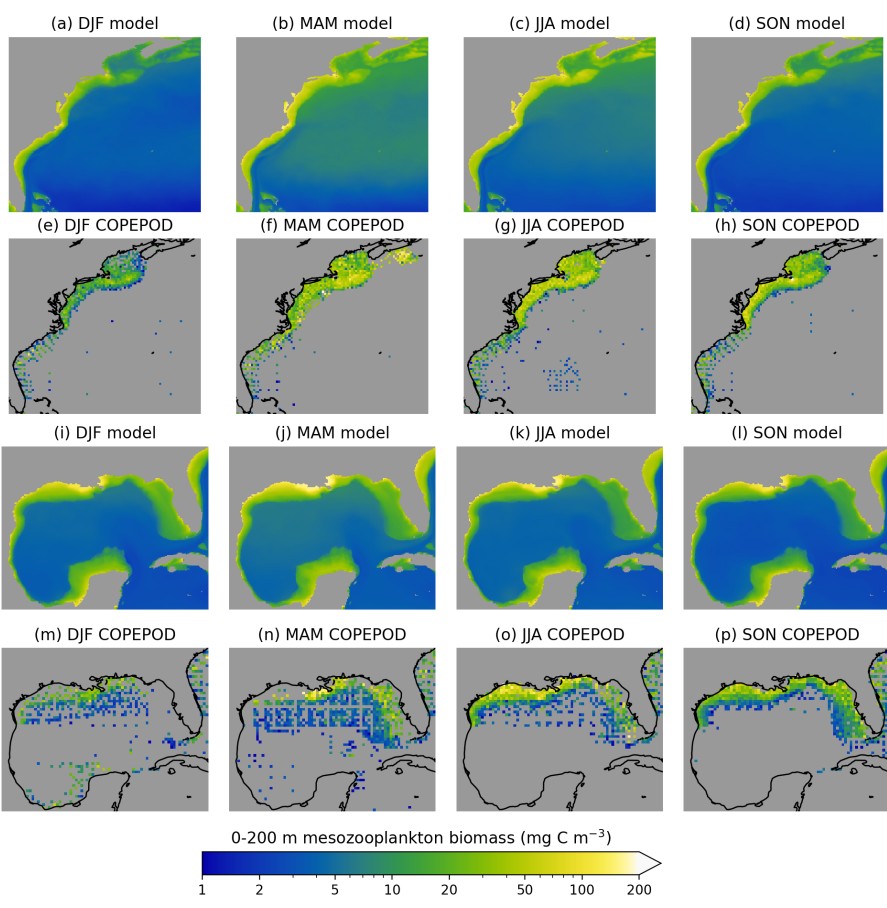

**Figure 14.** 0–200 m average mesozooplankton biomass climatology along the U.S. East Coast (a–h) and in the Gulf of Mexico (i–p). Observations from the COPEPOD dataset have been adjusted for approximate undersampling by scaling by a factor of 2.





Moving to the carbon system dynamics, the model annual average surface $pCO_2$ is a close match to the SOCAT and data-
derived products (Fig. 15a–f). The main bias relative to both observation-based products is higher modeled $pCO_2$ in the
subtropical gyre, a bias on the order of 10%. The model also appears to have biases in the Gulf of St. Lawrence and off
the coast of Newfoundland and Labrador; however, the observation-based products contain fewer observations in this region
and may be affected by seasonal ice cover (Landschützer et al., 2020). Comparing the seasonal variability of $pCO_2$ in the
model with the machine learning based estimate of Landschützer et al. (2020) (Fig. 15g–i), the model has higher variability
along the coast and south of the Gulf Stream path and lower variability in the southern Labrador Sea in the northeast corner of
the model domain. The bias towards higher seasonal cycle amplitude was also found in a $0.5°$ global simulation with COBALT
by Roobaert et al. (2022) and is likely attributable to the underestimation of the seasonality of non-thermal processes (i.e.,
biological update) in spring–summer and winter DIC mixing/supply in fall–winter.

Annual mean patterns in simulated alkalinity and dissolved inorganic carbon (DIC) generally agree with observation-based
climatologies, with spatial correlations of 0.93 and 0.87, respectively, and mean biases that are minor compared to the range
of values found across the domain (Fig. 16a–f). The primary alkalinity biases are a positive bias in the Northeast United States
and Canada, which is consistent with overly prominent high alkalinity/high DIC Gulf Stream waters. A low alkalinity bias is
evident in the southern parts of the model domain, particularly in the vicinity of large freshwater outputs in Central and South
America. This region is associated with a freshwater bias (Fig. 4), suggesting that the bias may arise from overly prominent
low alkalinity freshwater fluxes in this region. However, the observations in this region are highly uncertain: the GLODAPv2
dataset, which the Jiang et al. (2022) product sources from, contains almost no observations in the Caribbean and Gulf of
Mexico (Lauvset et al., 2021). The model DIC biases have a similar spatial pattern as the alkalinity biases, which is consistent
with these properties being driven by the same processes, although the magnitude of the DIC biases is less.

The modeled and observed surface aragonite saturation exhibit a similar negative gradient with latitude (Fig. 16g–i), and
the model is highly correlated with the observations ($r = 0.97$). While none of the mean values at the surface fall below a
saturation state, and only a few grid cells in nearshore estuarine areas fall below the value of 1.5 that is considered suboptimal
(Siedlecki et al., 2021), much of the northeastern coast of the United States and Canadian coastal waters fall below 2. In the
high alkalinity waters of the tropical North Atlantic, by contrast, the saturation state is often $> 4$. The largest discrepancies in
the model occur in the mid-Atlantic Bight, which tends to have a greater component of high $\Omega_{ar}$ tropical water and is consistent
with the stronger Gulf Stream influence noted previously, and in the Gulf of Mexico, where the observations may not sample
the nearshore region of low saturation state.

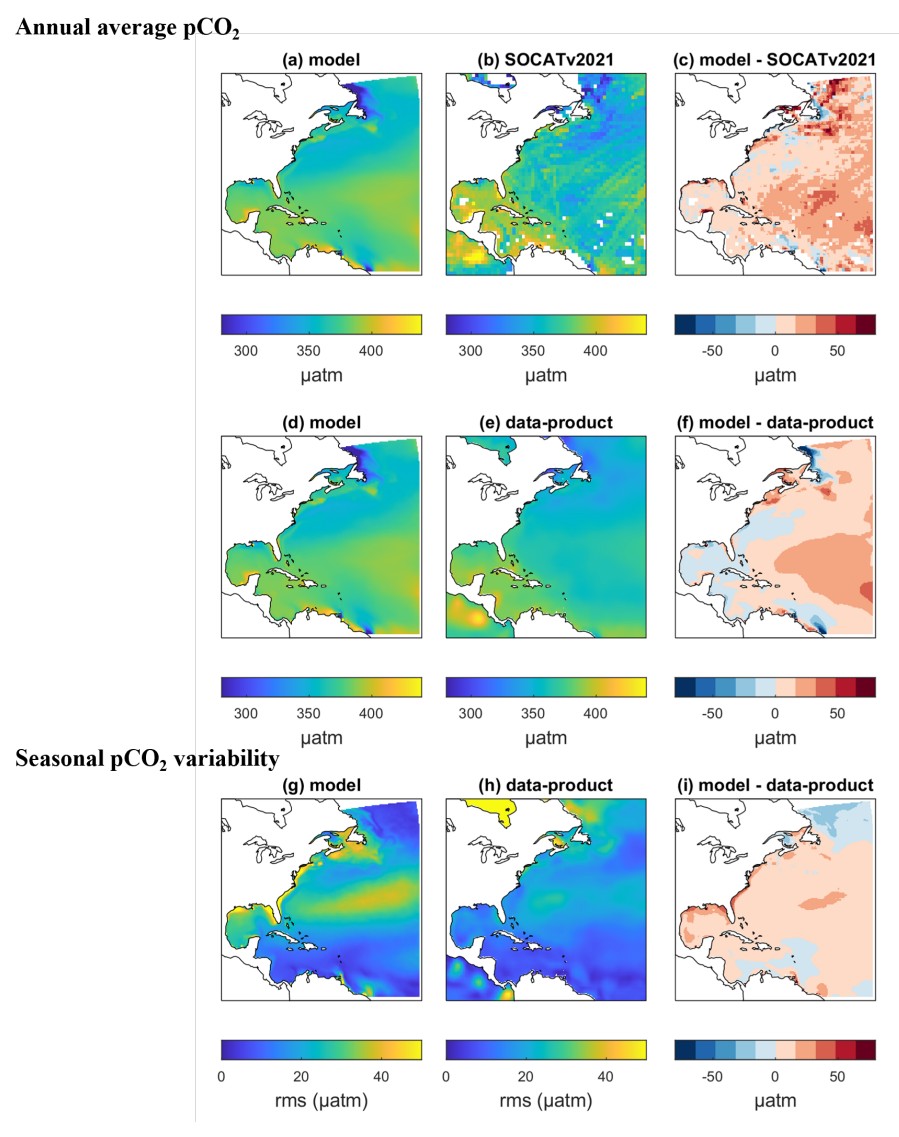

**Figure 15.** Modeled (left) and observed (center) surface $pCO_2$ spatial distributions (in $\mu$atm) and model bias (right) on annual average (a–f) and seasonal timescale (g–i). The seasonal timescale evaluation is performed on the seasonal amplitude which is expressed as the root mean square of the monthly climatology $pCO_2$ anomalies (rms, in $\mu$atm). The model evaluation is performed directly against $pCO_2$ observations derived from the Surface Ocean CO2 Atlas database version 2021 (SOCATv2021, Bakker et al. (2016)) and against a $pCO_2$ data-product generated from the SOCAT observation by a two-step neural network interpolation (data-product, Landschützer et al. (2020)).

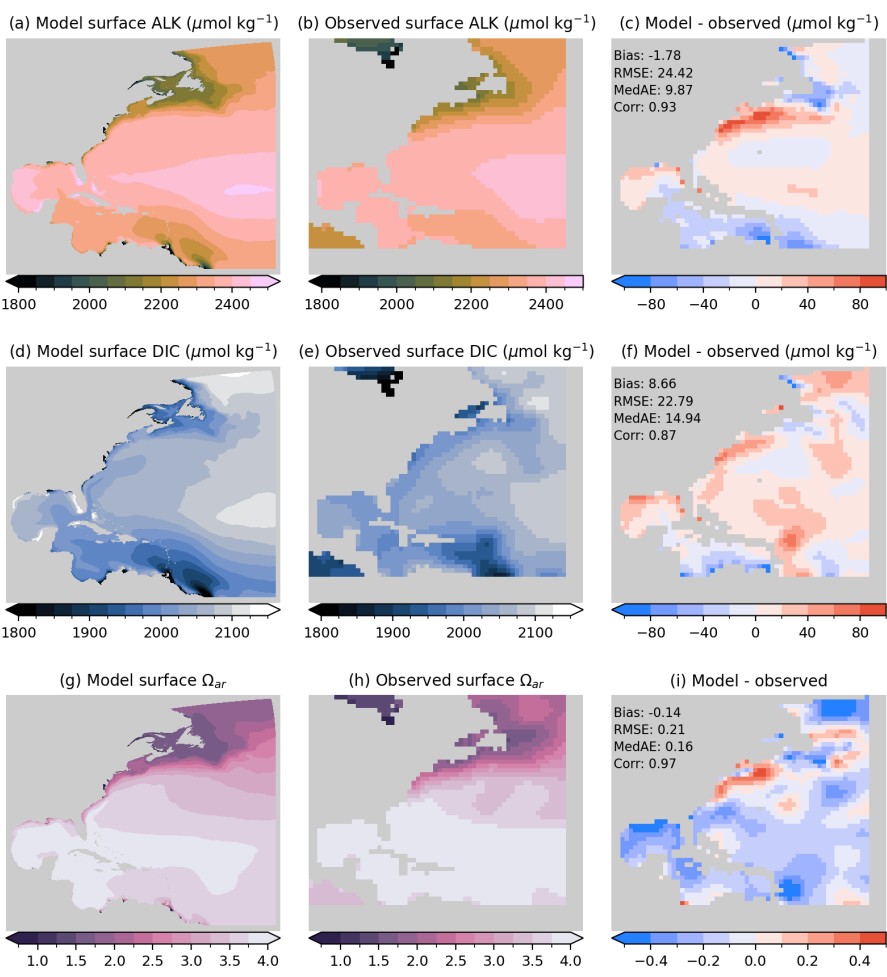

**Figure 16.** Mean surface alkalinity (a–c), dissolved inorganic carbon (d–f), and aragonite saturation state (g–i) in the model (left panels), the observation-derived climatology (center panels), and the difference between the model and observations (right panels).



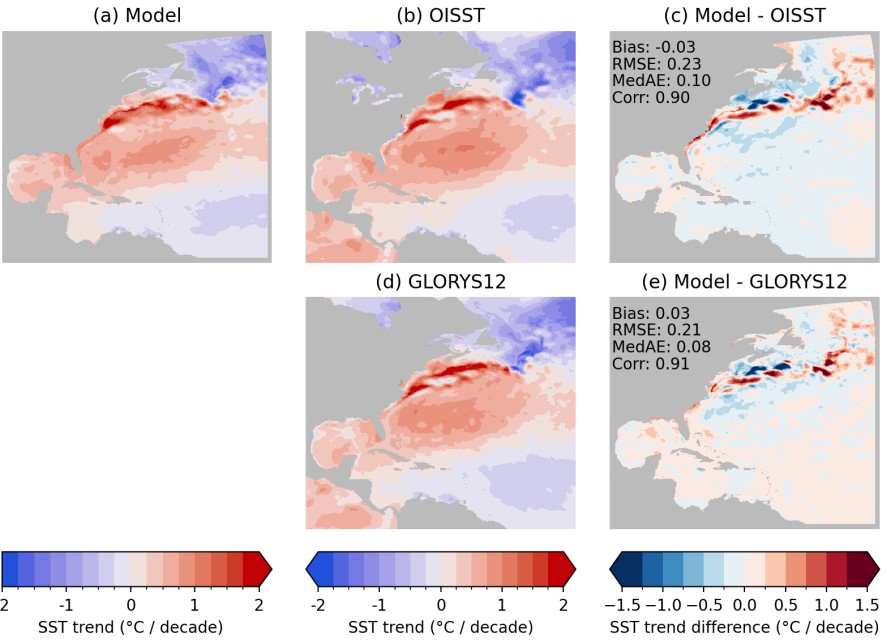

**Figure 17.** 2005–2019 sea surface temperature trends from the model (a) compared with trends from the OISST (b) and GLORYS12 (d) datasets, and the difference between the model and the datasets (c, e).

## 3.2 Relevant regional features and biogeochemical responses

As described in the introduction, one of the challenges for the intended applications of the modeling system described herein is to connect large-scale ocean dynamics to regional responses. This requires the model to capture both the large-scale patterns
(described in the preceding section), and more local marine resource relevant dynamics. One of the largest and most impactful changes in recent decades has been the substantial warming in the Northwest Atlantic Ocean, particularly along the Northeast U.S. coast. The model is largely able to simulate the observed linear SST trends over 2005–2019, with an overall spatial correlation of 0.90–0.91 and negligible mean bias (Fig. 17). Both OISST and the GLORYS12 reanalysis show a broad pattern of surface cooling at the southern edge of the subpolar gyre in the northwestern corner of the domain and surface warming
north of the Gulf Stream along the continental shelf and shelf break, which appears to be consistent with the predicted effect of a weakening Atlantic Meridional Overturning Circulation (Caesar et al., 2018), and this pattern is reproduced well in the model. However, there is some mismatch between the modeled and observed trends at fine scales along the shelf: the model temperature increase is too large along the Mid-Atlantic Bight and too small (although still positive) in the Gulf of Maine and Scotian Shelf and offshore of these regions beyond the shelf break.
Interannual bottom temperature variability within Northeast U.S. Ecological Production Units is simulated well by the model, although long-term trends are missed in some regions (Fig. 18). The correlation between the modeled and observed or reanalysis anomalies is highest in Georges Bank, where strong mixing couples the bottom temperature variability with the

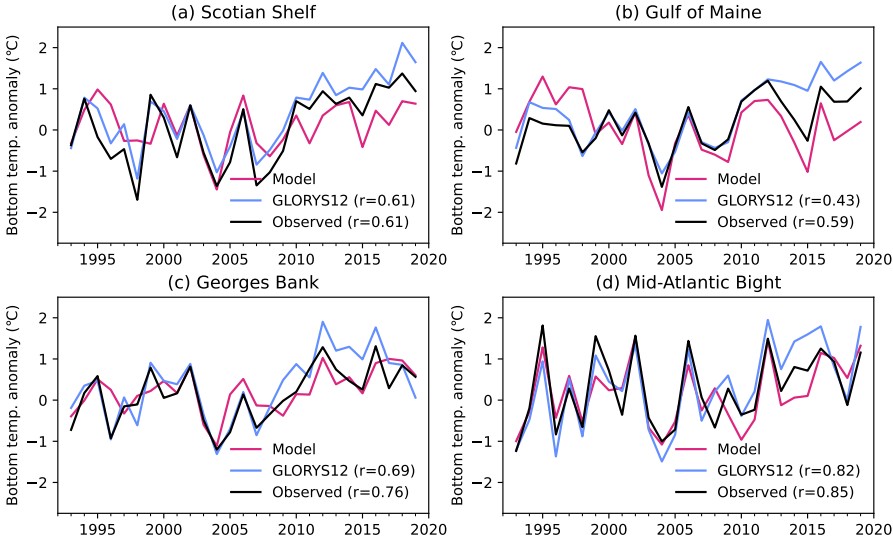

**Figure 18.** Annual average bottom temperature anomalies in four different Northeast U.S. ecological production units computed from model, reanalysis, and observed data.

atmospheric forcing, and the Mid-Atlantic Bight, and is lowest in the Gulf of Maine. Observations and reanalysis data from all EPUs show a general pattern of a sharp cold snap in 2004 that was associated with deep winter mixing (Taylor and Mountain, 2009) followed by a strong warming trend. Although the model reproduces these trends and the superimposed variability to some extent, it substantially underestimates the warming trend in the Gulf of Maine and the southwestern Scotian Shelf EPU. These underestimated bottom trends are consistent with the underestimated SST trends in these regions (Fig. 17). On the other hand, the observed and reanalysis time series began to diverge around 2010, with the reanalysis warming faster than the observations, and the model is generally closer to the observations than the reanalysis during this period.

The mean and range of modeled temperature and salinity at 150-200 m depth in the Gulf of Maine Northeast Channel is broadly consistent with the data from CTD and buoy observations and the GLORYS12 reanalysis (Fig. 19a). Relative to the reanalysis, the average model salinity is too high by 0.12, and the average model temperature is too warm by 0.25 °C. Compared to the analysis of of Saba et al. (2016), this warm bias is less than the warm bias of the 1/10° CM2.6 climate model, and substantially less than the warm bias of several degrees found in the 1/°4 CM2.5 climate model, which highlights the benefits of high-resolution downscaling. Although the reanalysis, CTD, and moored buoy data products all have slightly different means and ranges of temperature and salinity, all three products and the model show the same mixing of predominantly Warm Slope Water (WSW) and Labrador Slope Water (LSW) with a minor addition of Scotian Shelf Water. Compared to all three data products, however, the model temperature and salinity variability is confined within a smaller range. The model has relatively poor skill at simulating the time series of the mixing between these water masses (Fig. 19b–c); over the full 1993–2019 time period, the correlation between the model and data-derived water masses range from 0.30 to 0.43. Compared to the mooring-derived water masses, which are only available from 2004–2017, the model has much higher correlations of



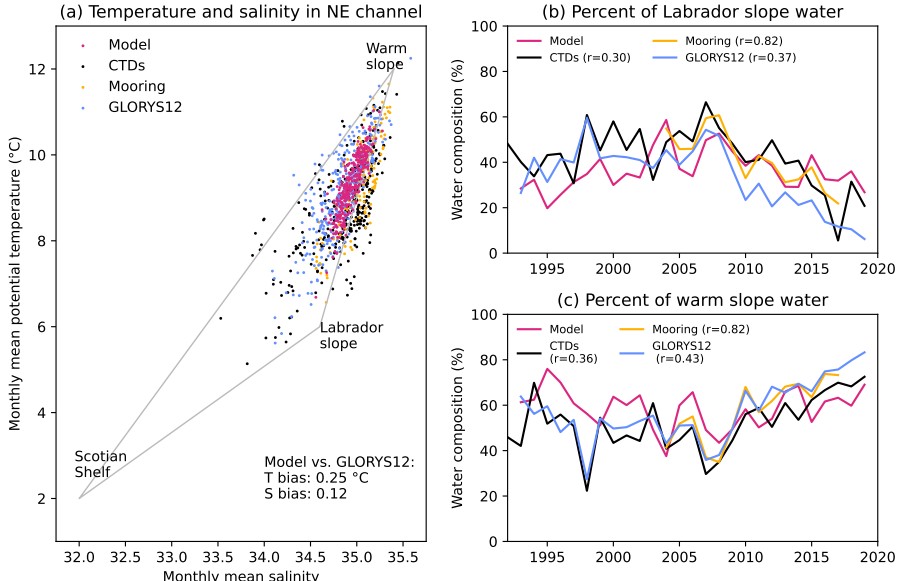

**Figure 19.** (a) Monthly potential temperature and salinity between 150–200 m within the Northeast Channel compared with known water masses. (b–c) Time series of the composition of the water between 150–200 m within the Northeast channel in terms of Labrador Slope Water (b) and Warm Slope Water (c).

0.82 for both LSW and WSW. This difference in correlation is due to the model missing several prominent fluctuations in water masses during the early part of the time series, including the significant drop in WSW in 1998, and accurately simulating many of the more recent fluctuations, including the second drop in WSW in 2007–2008 and the subsequent rebound. All three data

products show this rebound and a continued increase in WSW persisting through 2019, although the products diverge from each other to some extent during this time. The model also simulates some increase in WSW and decrease in LSW after 2010, although the changes are less pronounced than in any of the data products.

Model bottom temperatures in the Mid-Atlantic Bight Cold Pool region are biased warm by 0.70 °C on average (Fig. 20a–c), which is consistent with the warm bias seen in SST in this region. The model broadly simulates the spatial pattern of

temperature associated with the cold pool with a spatial correlation coefficient between the model and GLORYS12 reanalysis of 0.88. Year-to-year variability of temperature anomalies associated with the cold pool, as indicated by the index of du Pontavice et al. (2022) based on June–September average bottom temperature anomalies, also tracks the reanalysis data reasonably well (correlation coefficient between time series of 0.57) considering that this feature is typically challenging to model. This correlation is lower than the correlation of 0.85 between survey and model annual average bottom temperature anomalies across

the Mid-Atlantic Bight EPU (Fig. 18d), which is consistent with higher model skill during winter and spring when the water column is well mixed and bottom temperatures are connected to atmospheric forcing.

The model reliably simulates the spatial and temporal evolution of sea ice in the northwestern portion of the model domain (Figs. 21–22). The model has a modest bias towards earlier freezing and melting (Fig. 21), with more extensive sea ice coverage



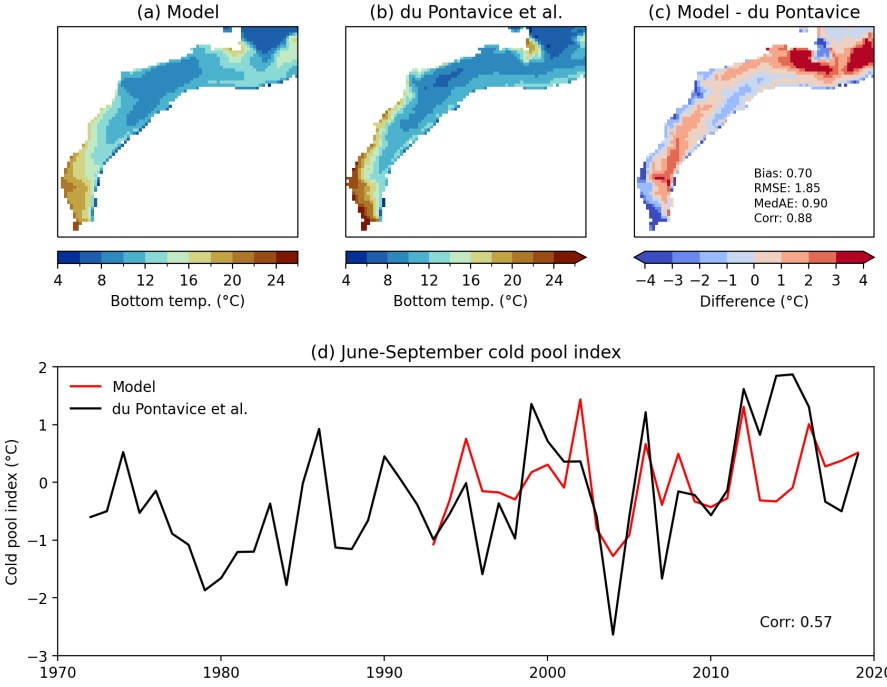

**Figure 20.** Bottom temperature from the model (a) and du Pontavice et al. (2022) (b) in the cold pool region averaged over June-September, and the difference between the two datasets (c). (d) Comparison of the model cold pool index, based on bottom temperature anomalies, with the same index calculated from regional ocean model (before 1993) and GLORYS reanalysis (1993–2019) data by du Pontavice et al. (2022).

than observed in December and January and less in April. The low bias in April may be due to the omission of open boundary

conditions for sea ice in the model, which does not allow the transportation of sea ice by the Labrador Current through the northern model boundary and into the domain. The time series of monthly sea ice extent (Fig. 22) shows that the model captures nearly all of the year-to-year variability in sea ice coverage, with correlation coefficients between 0.94 in January and 0.96 in February. The satellite data show an abrupt shift towards lower and more variable ice coverage beginning around 1995, and this shift is correctly simulated by the model (although the model simulation only began in 1993 and thus it is not certain whether

the model reproduces the relatively stable period before 1995).

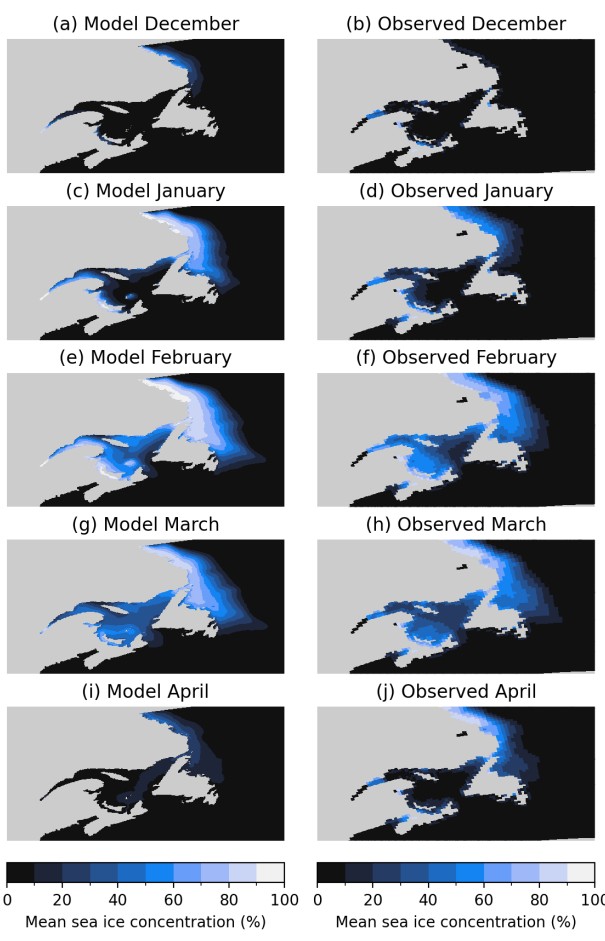

**Figure 21.** 1993–2019 monthly climatology of sea ice concentration from the model (left panels) and a satellite observation dataset (Cavalieri et al., 1996) (right panels).



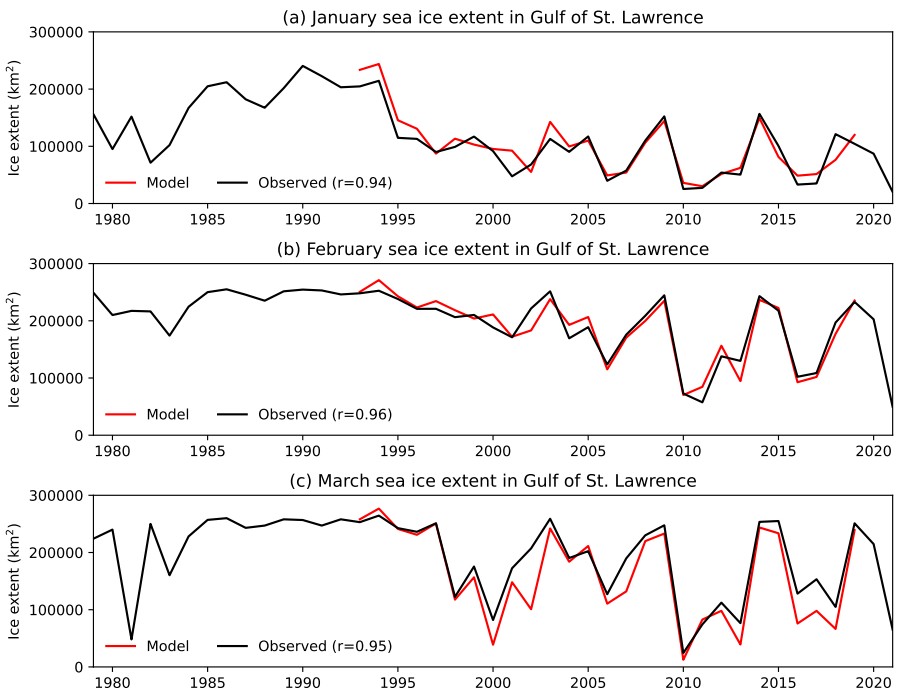

**Figure 22.** Time series of the extent of sea ice within the Gulf of St. Lawrence during January (a), February (b), and March (c) from the model and the satellite observation dataset (Cavalieri et al., 1996).



The model reproduces the primary response of surface chlorophyll in the Gulf of Mexico to both the El Niño and La Niña phases of ENSO, although the magnitude of the model response is not as large (Fig. 23a–d). In the satellite observations, surface chlorophyll is higher than normal along the majority of the Northern Gulf of Mexico coast during the winter and spring seasons of an El Niño event (Fig. 23a–d), although, as noted before, satellite estimates should be interpreted with caution in
this turbid nearshore region. The model simulates a positive chlorophyll response offshore of the Mississippi River delta and the western Gulf of Mexico, but fails to simulate the enhanced chlorophyll along the West Florida Shelf. In agreement with Gomez et al. (2019), the observed chlorophyll response during a La Niña winter and spring is opposite and slightly weaker compared to the response during El Niño (Fig. 23e–h). The model and remote sensing datasets generally agree that chlorophyll is lower than average over the majority of the region during La Niña winters. The model also reproduces increased surface
chlorophyll near the mouth of the Mississippi River and decreased chlorophyll along the Louisiana-Texas shelf during La Niña springs. However, the model again fails to simulate low chlorophyll anomalies along the West Florida Shelf and Southeast U.S. coast. Across the region plotted in Figure 23, the rank correlation between the model and satellite anomalies is slightly higher in spring than in winter for both El Niño and La Niña years.

Finally, bottom hypoxia along the Louisiana-Texas shelf is simulated reasonably well by the model (Fig. 24), especially
given the small scale over which this hypoxia occurs, the omission of extremely shallow areas in the model bathymetry (which has a minimum depth of 10 m), and the basic representation of coastal benthic processes in COBALT. The seasonal variation of hypoxic area in the model, peaking in July, is consistent with the area estimated from cruise observations and other data by Matli et al. (2020) during the months of May–September (Fig. 24a). In all months, the model hypoxic area is less than the observed area. During July, the model underpredicts hypoxic area by 3962 km$^2$ on average, or about 25% of the observed area
(Fig. 24b). About 1/4 of the interannual variability is correctly predicted by the model ($r = 0.51$).



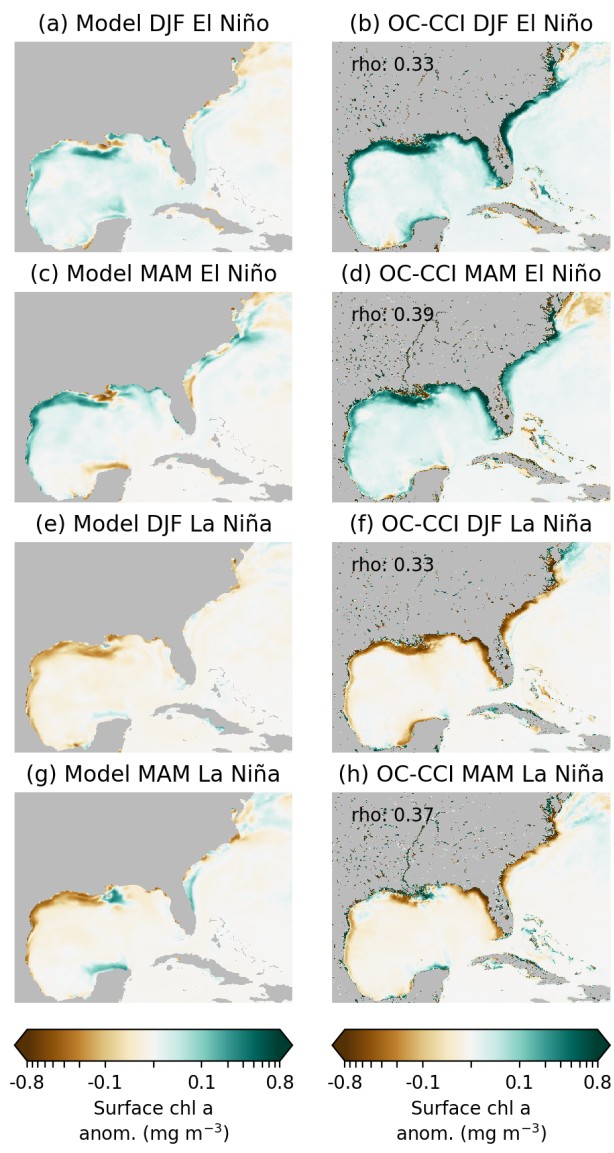

**Figure 23.** Winter and spring surface chlorophyll-*a* anomalies during El Niño years in the model (a, c) and satellite estimates (b, d), compared with the same during La Niña years (e–h). To display the wide range of chl-a anomalies, values within $\pm0.1$ mg m$^{-3}$ are shaded on a linear scale, while values outside of this range are shaded on a logarithmic scale. Annotated correlation values give the Spearman rank correlation between the model and satellite anomalies over the plotted region.





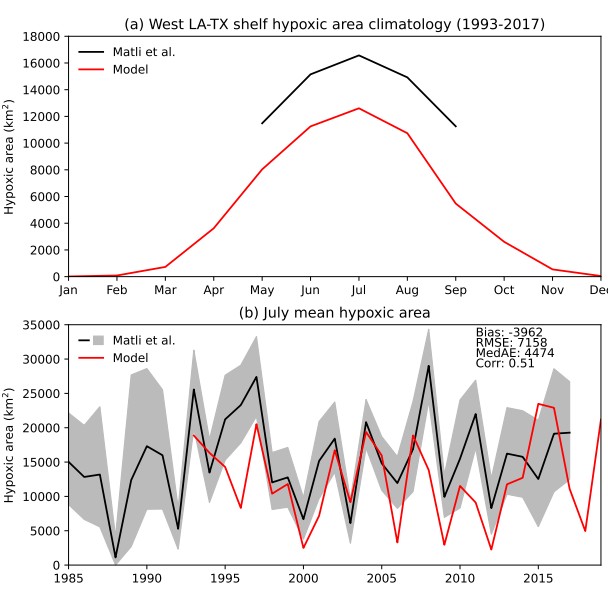

**Figure 24.** Monthly climatologies of hypoxic area (area with a July mean bottom oxygen concentration below 2 mg l$^{-1}$) over the LA-TX shelf from the model and geostatistical estimates from Matli et al. (2020) (a) and time series of July mean hypoxic area from the model and Matli et al. (2020) (b). Gray shading in (b) denotes the 95% confidence interval for the Matli et al. (2020) data.





## 3.3 Computational performance

In Figure 25, we evaluate the total run time of the model under several choices of grid decomposition to determine whether the computational cost of the model is low enough to enable running the large ensembles of long simulations that are needed to support the intended applications. Using the 40x40 decomposition, which distributes the 775x845 model grid onto a 40x40 grid of processing elements (PEs), the model can run one year of simulation in about 11.5 hours of wall clock time. As the number of PEs is increased, the wall clock time is reduced, but with an efficiency that is significantly less than 1: the 50x50 decomposition has an efficiency of just under 0.8 (in other words, the total computational cost in processor-hours of the 40x40 case is just under 0.8 times the cost of the 50x50 case), and the 70x70 decomposition has an efficiency of less than 0.6. Despite the increased total computational costs of the larger PE layouts, these layouts support run times fast enough meet our informal performance criterion of 3 simulation years per day. Both the 60x60 and 70x70 cases exceed this threshold, which allows them to run nearly a century of continuous simulation in one month (assuming no cluster queue wait time). The 50x50 configuration, which was used for the main hindcast simulation in this paper, nearly meets this goal as well. The 40x40 configuration, although slower, can still complete two simulation years per day, which allows the completion of two 30-year time slices (for example, a historical simulation and a future projection) in one month. This configuration is also optimal for reducing resource usage during individual hindcast simulations, where the results can be continuously assessed while the model runs, or for forecast experiments where multiple ensemble members can be run efficiently in parallel. Furthermore, the scaling efficiency of this tracer-heavy model is better than seen in MOM6 models without coupled biogeochemistry, which generally have a practical limit of 15x15 points per PE due to the 2-dimensional barotropic solver.

A key feature of MOM6 that improves the computational economy is the ability to efficiently integrate the thermodynamics and tracer processes on a longer time step than the baroclinic and barotropic dynamics time steps that are highly limited by stability concerns. Because the coupled COBALT biogeochemical model integration is called during the thermodynamics time step and COBALT introduces 40 new tracers that must be advected and diffused by the ocean model component during the thermodynamics time step, taking longer thermodynamics time steps can substantially reduce the total computational cost of the model. In the present configuration, we use an 1800 second thermodynamics time step, which is 3 times longer than the 600 second baroclinic time step. To evaluate the cost savings of this feature, we ran two experiments with a 600 second thermodynamics time step and plotted the run times as empty shapes in Figure 25. For a given grid decomposition, an 1800 second thermodynamics time step results in a factor of 2 decrease in the total run time compared to a 600 second thermodynamics step. Because of inefficiencies in scaling with large numbers of PEs, for a given total run time, the reduced number of PEs needed by the longer thermodynamics time step results in a factor of 3 decrease in the computational cost (compare the 40x40 decomposition with an 1800 s step with the 70x70 decomposition with a 600 s step). MOM6 is also computationally efficient at handling tracers in general: using the 1800 s thermodynamics step, the inclusion of coupled COBALT biogeochemistry and its 40 tracers only results in about a 2.5 times increase in the computational cost (not shown).



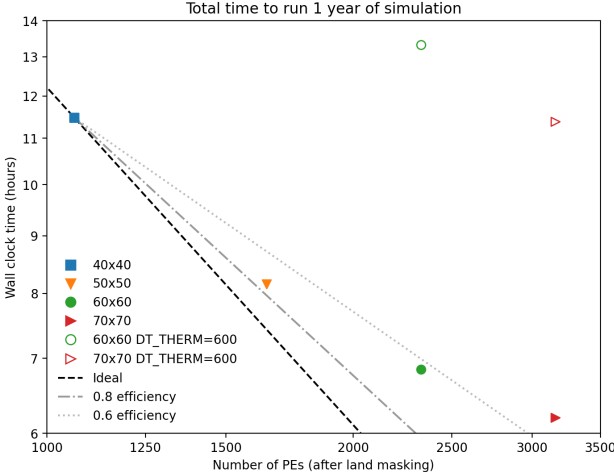

**Figure 25.** Total wall clock time needed to run one year of model simulation as a function of the number of processing elements (PEs). Shapes and colors indicate the grid of PEs onto which the model domain is distributed before eliminating all-land PEs. Dotted and dashed lines indicate various scalings relative to the 40x40 case. The two empty shapes are simulations with the thermodynamics time step set to the same as the dynamics time step. Both plot axes are on a logarithmic scale.

## 4 Discussion

MOM6-COBALT-NWA12 exhibited good performance at simulating a wide range of ecosystem-relevant physical and biogeo-
chemical diagnostics while also satisfying computational performance constraints. Long-term means and seasonal climatolo-
gies were simulated well for nearly all metrics, with the main biases across quantities often related to challenges in simulating
the Gulf Stream. The 1/12° resolution used here is only slightly higher than the 1/10° resolution considered to be a necessary
but not sufficient condition for resolving the Gulf Stream separation and path (Chassignet and Marshall, 2008). A version of
the model with highly refined resolution, on the order of 1/25° or 1/50°, may improve the Gulf Stream path and variability and
reduce the biases, as has been found in studies of other models on similar domains (Chassignet and Xu, 2017, 2021). However,
such increases in resolution would also greatly increase the computational costs. An 8 times increase in computational cost
is generally incurred for each doubling of the horizontal grid resolution (4 times for the increase in the number of grid cells,
and 2 times for the required time step decreases to maintain numerical stability, although in MOM6 the thermodynamics time
step need not be decreased with increasing resolution and this factor may be more favorable). We are currently running higher
resolution simulations to better quantify the model skill and computational cost tradeoffs associated with enhanced resolution
in the NWA12 domain.

Metrics that evaluate the model on the basis of interannual to decadal variability, rather than long-term means, also showed
significant skill, particularly for lower-frequency variations, along with some errors possibly related to the Gulf Stream bias.
Variability of the Gulf Stream position, bottom temperatures in the Northeast U.S. EPUs, sea ice in the Gulf of St. Lawrence,
and sea surface temperature trends all compared reasonably well with the respective observed time series. However, the model





slightly underestimated recent surface warming in the Gulf of Maine and largely failed to simulate the recent bottom warming, and the composition of deep water in the Northeast Channel was only moderately coherent with observations during the full 1993–2019 time period, though agreement with buoy data over the last 15 years was stronger. Some of the lower performance in this region may be caused by unpredictable stochastic variability, such as warm core eddies shed from the Gulf Stream that bring warm and salty water to the shelf and coast (Du et al., 2021; Gawarkiewicz et al., 2019). While eddies are present at 1/12° resolution, the formation and evolution of individual eddies is not deterministically simulated without data assimilation. This is consistent with the lower correlation with observations for the monthly Gulf Stream index compared to the 25-month rolling mean.

During development of the model, we found several parameters that exerted a strong influence over the physical properties of the model simulation. First, the separation point and path of the Gulf Stream was influenced by the biharmonic viscosity parameters: higher viscosity produced a later, northerly separation, and lower viscosity produced an earlier, southerly separation. This result is consistent with numerous other studies using other models (e.g., Bryan et al., 2007; Chassignet and Garaffo, 2001). Our biharmonic viscosity parameterization, the maximum of a velocity scale of 1 cm s$^{-1}$ times the grid spacing cubed and a Smagorinsky viscosity with a coefficient of 0.015, results in a typical viscosity of about $3.8 \times 10^9$ m$^4$ s$^{-1}$ at the latitude of Cape Hatteras. The viscosity cannot be substantially reduced at this resolution without introducing grid scale noise and instabilities or increased without shifting the Gulf Stream separation northward and exacerbating the warm temperature biases along the shelf. Compared to other studies, the viscosity in the MOM6-NWA12 is lower than the viscosity in the 0.1° ocean resolution models of Li et al. (2022) and Sasaki et al. (2020). The velocity scale of 1 cm s$^{-1}$ is the same as the 1/4° and 1/2° models of Adcroft et al. (2019), although A19 used a higher Smagorinsky coefficient of 0.06 and included an additional Laplacian viscosity in regions of the 1/2° model where the resolution was coarser than the first baroclinic deformation radius. Chassignet and Garaffo (2001) and Chassignet and Xu (2017) also used the same biharmonic velocity scale; however, Chassignet and Garaffo (2001) found that a small biharmonic viscosity alone produced an early separation of the Gulf Stream and could only obtain an accurate simulation with Laplacian viscosity added everywhere in the model domain. Aside from viscosity, MOM6-NWA12 has other differences from these models, including explicit tides and z* vertical coordinates rather than isopycnal or hybrid coordinates, and identifying which factors contribute to obtaining a satisfactory simulation with lower viscosity is left for future work. This line of research is likely to benefit from the flexible vertical coordinate capability of MOM6.

Second, the front length specified in the parameterization for restratification by submesoscale eddies has a strong control over the mixed layer depth in MOM6, as noted by Adcroft et al. (2019). The front length of 1500 m used here is greater than the 500 m used in Adcroft et al. (2019), which results in a decreased restratification effect. The front length is also greater than suggested by the scaling developed by Bodner et al. (2023), which predicts front lengths shorter than even 500 m throughout most of the NWA12 model domain. Shorter front lengths would decrease mixed layer depths, which would be beneficial in some regions of the domain that have a bias towards deeper winter mixed layers (Fig. 5). However, the reduced mixing caused by a stronger restratification effect would have other impacts, including increasing the bottom temperature bias and reducing the extent of the cold pool in the Mid-Atlantic Bight (Fig. 20). The chosen value of 1500 m is thus a reasonable compromise.



Finally, the coefficient used in the scalar approximation of the effect of tidal self-attraction and loading (SAL) has a strong control on tidal range in the model, particularly in the Gulf of Maine and Gulf of St. Lawrence. In this scalar SAL approximation, the effect of SAL is equal to a single, constant coefficient times the local model free surface elevation. The coefficient value of 0.01 used in this study is significantly lower than values ranging from 0.085–0.12 commonly used in global tide models (Accad and Pekeris, 1978; Ray, 1998; Stepanov and Hughes, 2004). In the NWA12 regional model, increasing the SAL coefficient increases M2 amplitude in the Gulf of St. Lawrence and decreases it in the southwest Gulf of Maine. Increasing SAL above 0.01 would thus reduce the Gulf of Maine M2 amplitude bias in the current model but increase it in the Gulf of St. Lawrence (Fig. 8). The value of 0.01 is a compromise between these two biases and the effects they have on regional hydrodynamics. In reality the effect of SAL is not constant but varies as a function of the tidal spatial scales (which in turn vary with water depth) and other factors (Ray, 1998; Stepanov and Hughes, 2004), and modeling studies have found that the coefficient should be smaller than the typical 0.085–0.12 over most of the Atlantic Ocean, and particularly in the Gulf of Mexico and along the U.S. East Coast (Irazoqui Apecechea et al., 2017; Stepanov and Hughes, 2004). We consider the scalar value of 0.01 to produce results that are sufficiently accurate for the intended use of the current version of the model, but separate ongoing studies may explore the effect of more sophisticated models of SAL and the impacts on coastal hydrodynamics.

## 5 Conclusions

As a first step towards the goal of providing information about historical ocean-ecosystem conditions and possible future changes that could support living marine resource management and applications, we developed a regional model of ocean dynamics and biogeochemistry for the Northwest Atlantic Ocean. Comparison of a model historical simulation with data from reanalysis and observations showed that the model generally performed well at simulating both historical mean conditions and significant trends and variability with minimal drift. This suggests that the model can provide the accurate information needed for many applications; notably, several of the evaluation metrics in this paper and the observations for them were taken directly from reports used to inform fisheries managers. However, more detailed skill evaluations specific for each intended application will likely be necessary to better understand the reliability and uncertainty of the model simulations.

Not all aspects of the model simulation reproduced the observations as well as might be desired. Interannual variability of fine-scale features, including the Mid-Atlantic Bight cold pool and the area of hypoxia along the LA-TX shelf, were simulated with correlation coefficients around 0.5, which is commonly considered a lower bound for useful prediction skill. Recent warming at the surface and bottom of the Gulf of Maine was also underestimated. Future model simulations focused more specifically on these regions may benefit from employing additional downscaling by nesting a 1/25° or higher local model within the 1/12° regional NWA12 domain to obtain some of the benefits of higher resolution with a lower computational cost. Adjustments to the spacing of the vertical coordinates to enhance resolution in coastal areas, or switching from z-level to an adaptive hybrid coordinate that is locally terrain following, may also improve the simulation of these features. Nested local domains would also allow adjusting the model parameters, such as the mixed layer eddy front length and tidal self-attraction and loading coefficient, to optimal values for each region. Furthermore, as it was not possible in this paper to compare the model





with every relevant metric and dataset within the entire NWA12 domain, development of local domains could focus on a more
comprehensive evaluation for the local area. However, unless two-way nesting was employed, these smaller, nested domains
would limit the applicability for large-scale issues like cross-boundary shifts in species distributions that may occur under
climate change, and a proliferation of different models could introduce confusion and complexity and complicate development
of the core model components.

Although historical model simulations can provide useful information for LMR applications, for example by reconstructing
historical conditions when observations were sparse (du Pontavice et al., 2022), forecasts and projections of future conditions
are essential for managing and adapting to the impacts of climate variability and change on LMRs (Tommasi et al., 2017b).
Interdisciplinary research has repeatedly emphasized that forecast users benefit from being provided with information about
forecast accuracy and uncertainty (e.g., Ramos et al., 2013; Roulston et al., 2006). However, providing reliable uncertainty
information generally requires running long ensemble simulations, and the imposing computational costs of these simulations
has limited the number of cases where this information is provided (Lewis et al., 2022). The model developed here takes
advantage of the efficiency of MOM6 to achieve runtimes of more than 3 simulation years per day, our informal goal for a
useful model that can run large ensembles in reasonable times. Although there are areas where this first version of the model
can be improved, we believe it forms a useful starting point for following the recommendations of Dietze et al. (2018) by
iteratively creating testable predictions that can both inform LMR applications and produce insights on how to improve the
890 model and make better predictions.

*Code availability.* The source code for each component of the model has been archived at https://doi.org/10.5281/zenodo.7893349. MOM6 is
built on an open development paradigm, and the Git repositories at https://github.com/mom-ocean/MOM6 and https://github.com/NOAA-GFDL/
MOM6 provide a means for the community to obtain updated and experimental source code, report bugs, and contribute new features. Repos-
itories for the other model components are also available at https://github.com/NOAA-GFDL.

*Data availability.* All model output that was analyzed in this paper has been published at https://doi.org/10.5281/zenodo.7893387. Model
parameter files and prepared forcing files are published at https://doi.org/10.5281/zenodo.7893727.

The datasets used for comparison with the model and the URL or DOI where the data can be downloaded are listed as follows. GLO-
RYS12 reanalysis (https://doi.org/10.48670/moi-00021); OISST v2 (https://psl.noaa.gov/data/gridded/data.noaa.oisst.v2.highres.html); Re-
gional temperature and salinity climatologies (https://www.ncei.noaa.gov/products/regional-ocean-climatologies); Mixed layer depth (https:
900 //doi.org/10.17882/91774); Global ocean gridded sea surface heights (https://doi.org/10.48670/moi-00148); TPXO9 (https://www.tpxo.net/
home); OC-CCI v6.0 (https://www.oceancolour.org/); COPEPOD (https://www.st.nmfs.noaa.gov/copepod/biomass/biomass-fields.html); World
Ocean Atlas 2018 (https://www.ncei.noaa.gov/archive/accession/NCEI-WOA18); SOCATv2021 (https://doi.org/10.25921/yg69-jd96); Sur-
face pCO2 climatology (https://doi.org/10.25921/qb25-f418); Alkalinity, DIC, and aragonite saturation (https://doi.org/10.25921/g8pb-zy76);
EPU bottom temperatures (https://github.com/NOAA-EDAB/ecodata/releases/tag/3.0); Northeast U.S. CTD profiles (ftp://ftp.nefsc.noaa.





gov/pub/hydro/matlab_files/yearly); Buoy N01 (http://www.neracoos.org/erddap/info/index.html?); Sea ice concentration (https://doi.org/ 10.5067/MPYG15WAA4WX); Cold pool index (https://github.com/NOAA-EDAB/ecodata/releases/tag/3.0).

The datasets used to create the model forcing and the URL or DOI where the data can be downloaded are listed as follows. GLO-RYS12 reanalysis (https://doi.org/10.48670/moi-00021); TPXO9 (https://www.tpxo.net/home); World Ocean Atlas (https://www.ncei.noaa.gov/archive/accession/NCEI-WOA18); GloFAS (https://doi.org/10.24381/cds.a4fdd6b9); USGS Gauge 07374525 (https://waterdata.usgs.gov/monitoring-location/07374525/); ERA5 (https://doi.org/10.24381/cds.adbb2d47); Carter et al. (2021) alkalinity and DIC estimation algorithm (https://doi.org/10.5281/zenodo.5512697); RC4USCoast (https://doi.org/10.25921/9jfw-ph50); GlobalNEWS2 (https://doi.org/10.1016/j.envsoft.2010.01.007); Meinshausen et al. (2017) atmospheric $CO_2$ (https://doi.org/10.22033/ESGF/input4MIPs.1118, https://doi.org/10.22033/ESGF/input4MIPs.9866). Data from Lavoie et al. (2021) and Stock et al. (2014) can be obtained by contacting the corresponding authors.

## Appendix A

**Table A1.** Parameter values used for the 4 phytoplankton group COBALT formulation enlisted herein. The rationale underlying the variation of parameters across sizes and functional types is consistent with Stock et al. (2020), and further discussion can be found therein.

| Param | Name | Units | Small | Medium | Large | Diazo |
|---|---|---|---|---|---|---|
| $P_{\max}^C$ | Maximum photosynthetic rate at 0 °C | day$^{-1}$ | 0.9 | 1.0 | 0.9 | 0.6 |
| $\alpha^{\text{chl}}$ * | chl-a-specific initial slope of the photosynthesis-light curve | g C (g chl)$^{-1}$ ($\mu$mol photons m$^{-2}$)$^{-1}$ | $2.5 \times 10^{-5}$ | $1.25 \times 10^{-5}$ | $0.5 \times 10^{-5}$ | $0.5 \times 10^{-5}$ |
| $\theta_{\max}$ | Maximum chl-a:C ratio | g chl (g C)$^{-1}$ | 0.035 | 0.045 | 0.055 | 0.035 |
| $k_{\text{no3}}$ | Half-saturation for nitrate | $\mu$M | 0.5 | 1.0 | 2.5 | 5.0 |
| $k_{\text{nh4}}$ | Half-saturation for ammonium | $\mu$M | 0.01 | 0.02 | 0.05 | 0.1 |
| $k_{\text{po4}}$ ** | Half-saturation for phosphate | $\mu$M | 0.01 | 0.02 | 0.05 | 0.1 |
| $k_{\text{fed}}$ | Half-saturation iron update | $\mu$M | 0.0004 | 0.0008 | 0.002 | 0.004 |
| $k_{\text{fe2c}}$ | Half-saturation for iron cell quota | mol Fe (mol C)$^{-1}$ | 2 | 4 | 10 | 12 |
| fe2c$_{\max}$ | Maximum Fe:C ratio | mol Fe (mol C)$^{-1}$ | 50 | 250 | 500 | 500 |
| N:P$_{\min}$ | Minimum N:P ratio (P-replete) | mol N (mol P)$^{-1}$ | 20 | 16 | 14 | 40 |
| sink$_{\max}$ | Maximum sinking rate (non-aggregated) | m day$^{-1}$ | 0 | 1 | 5 | 1 |
| $m_{\text{agg}}$ | Aggregation loss rate constant | day$^{-1}$ $\mu$mol N$^{-1}$ kg | 0.05 | 0.10 | 0.25 | 0 |
| $m_{\text{vir}}$ | Viral loss rate constant | day$^{-1}$ $\mu$mol N$^{-1}$ kg | 0.25 | 0.125 | 0.05 | 0.05 |

∗Values in $\mu$mol photons m$^{-2}$ s$^{-1}$ were converted to Watts m$^{-2}$ assuming 4.60 Watts m$^{-2}$ per $\mu$mol photons m$^{-2}$ s$^{-1}$ (Kirk, 1994). ∗∗Value scaled by N:P / N:P$_{\max}$ to account for P-frugality.





**Table A2.** The innate prey availability applied for each zooplankton consumer (column) to each potential prey item (column). The maximum innate availability is 1. Prey switching between herbivory and carnivory is parameterized as described in Stock et al. (2008).

| Predator | Prey | | | | | | | |
|---|---|---|---|---|---|---|---|---|
| | B | SP | MP | LP | DIAZO | SZ | MZ | LZ |
| SZ | 0.5 | 1.0 | 0.4 | 0 | 0 | 0 | 0 | 0 |
| MZ | 0 | 0.4 | 1.0 | 0.25 | 0.75 | 1.0 | 0 | 0 |
| LZ | 0 | 0 | 0.4 | 1.0 | 0.4 | 0 | 1.0 | 0 |

*Author contributions.* ACR, CAS, AA, RH, MJH, KH, NZ, RD, WC contributed source code for regional MOM6, COBALT, SIS2, and/or other components of the model framework. ACR, CAS, EJD, FG, DK, DL, JS contributed to preparation of model input files. ACR, CAS, EC, MA, HdP, FG, JGJ, DL, LR, AR, VS, SS contributed to evaluation and interpretation of the model results. ACR and CAS prepared the initial draft of the manuscript. All coauthors participated in discussions during various stages of the model development and evaluation and read and approved the final version of the manuscript.

*Competing interests.* The authors declare that they have no conflict of interest.

*Acknowledgements.* Vimal Koul and John Krasting are acknowledged for their helpful comments and suggestions on an internal review of the manuscript. Advice from Mitch Bushuk on the remotely sensed sea ice data and suggestions from Sang-Ki Lee are also greatly appreciated. ACR, EC, DK, JS, LR, SS, and SS were partly funded by grants from NOAA's Climate Program Office. EC, KH, JS, FG and WC were partly funded by NOAA's Climate Portfolio. EC, KH, JS, and DK were partly funded by a grant from the Cooperative Institute for Modeling the Earth System. This study has been conducted using E.U. Copernicus Marine Service Information: https://doi.org/10.48670/moi-00021, https://doi.org/10.48670/moi-00148.





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
