# Peer review of "A high-resolution physical-biogeochemical model for marine resource applications in the Northwest Atlantic (MOM6-COBALT-NWA12 v1.0)"

_Geoscientific Model Development, 2023_

## Author Comment (AC1)

**Summary:**

We thank the reviewer for providing a detailed and well-organized set of suggestions to improve the manuscript. Below, we have restated each review comment in bold text followed by our reply.

**General comments:**

**1. Water masses – Vertical sections. The comparison metrics used are very informative. However, in my opinion, it will be helpful to add some comparisons between the model and observations/reanalysis in terms of key vertical sections (or profiles) to better demonstrate the representation of different water masses in the model. For example, I suggest adding such comparisons for the winter and summer temperature and salinity climatologies for either: (i) sections of depth versus distance form the coast to beyond the shelf break, for maybe the Gulf of Maine+Georges Bank+slope, the MAB+shelf break and the Gulf of Mexico; or (ii) representative vertical profiles (perhaps averages) over the 4 EPUs. Consider doing the same for phosphate (or nitrate) and perhaps DIC.**

We have inserted a new figure (now Figure 18) that shows vertical profiles of seasonal temperature climatologies for the 4 EPUs. We added only temperature because it has stronger seasonal variation than salinity, and we don't have reliable climatologies of biogeochemical variables for these regions (World Ocean Atlas would have data on these variables, for example, but it is not accurate at these fine coastal scales).

[Figure]

**2. Main circulation features. In my opinion, it would beneficial to have a visualisation of the model mean circulation. I suggest adding a figure/map that compares the surface (or the vertical averaged) currents in the model to the reanalysis. The authors can decide how best to visualise this (perhaps using arrow plots with a background colormap for speed).**

We have included below a figure of the model mean surface current speed and direction. This figure shows that the model includes the well-known large-scale features, including the Loop Current, Gulf Stream, inflowing Labrador current, and counterclockwise circulation around the Gulf of Maine. One feature that appears poorly simulated is the Mid-Atlantic Bight shelfbreak jet.

[Figure]

We have not added this visualization to the manuscript for several reasons. First, the geostrophic currents can already be visually approximated from the evaluation of mean sea surface height. Second, quality long-term observations to compare to are lacking. Even the GLORYS12 reanalysis does a poor job at reproducing currents near the coast; for example, it does not have the cyclonic circulation in the Gulf of Maine. Third, during the model simulation we saved fewer outputs for velocity, and some of these diagnostics like the mean speed are affected by the presence of tides. For the figure above, for example, we had to use monthly mean surface u and v velocity components, which will bias the results towards lower average speeds.

**3. Streamlining (this is just a suggestion): The introduction is very informative; however, in my opinion, the discussion for the ocean conditions and variability in the North West Atlantic (lines 24-90) is a little too descriptive/long and could be condensed to better highlight the shortcomings due to limited availability of skilful high-resolution regional predictions and projections. I suggest streamlining lines 24-90, but this is just a suggestion and up to the authors' preference.**

We appreciate the suggestion. We hope to follow this manuscript with studies on applications of the model, as well as eventual papers on continued model development and improvement. We

have thus elected to leave the introduction as is, with the intention to provide a fairly comprehensive overview that sets the motivation for the present and future papers and can be referred to in the future.

**4. Figures and latitude longitude (this is just a suggestion): I understand that the map-figures do not have latitude and longitude so as to preserve space and make them look more compact. However, adding latitude and longitude to all of the map-figures would help readers who are not familiar with the region to easily follow the features shown in these figures; particularly in figures that involve zooms in specific regions (e.g., Figures 6,13, 14, 20, 21, 23).**

We have added lat/lon ticks and labels to all of the figures that zoom in to specific regions (Figures 6, 7, 13, 14 and 20, 21, 23 (now 21, 22, 24)).

**5. Seasonal mean estimates. The seasons for temperature, and chlorophyll (Figures 3, 12, 23) are defined as: (i) December-February, (ii) March-May, (iii) June-August, (iv) September-November; while the seasons for nutrients (Figures 10 and 11) are defined as: (i) January-March, (ii) April-June, (iii) July-September, (iv) October-December. I was wondering if there is a reason for using different months to define the seasons for nutrients. If so, please explain it in the text. If not, it might be better to keep the definition of the seasons consistent for all the fields.**

The different definition of the seasons used in the nutrient figures originates from the World Ocean Atlas dataset. WOA provides data in seasons starting with JFM, whereas in most cases we prefer to use the standard meteorological seasons starting with DJF. (World Ocean Atlas does provide a monthly climatology, which could be averaged to standard seasons; however, considering the sparseness of the observations in some areas,we prefer to use the WOA seasonal climatology which will have more data available for the WOA interpolation and smoothing steps). We have not changed the analysis, but have now highlighted in the text the different definition of the seasons and the reasoning why.

**Specific comments:**

**6. Line 133. I am not sure what "… coastwide extend to address the prominent cross-boundary issue expected under climate change" means here. Maybe it refers to along-shelf propagating signals, such that for example you have a large domain covering the whole US East coast? If yes, in my opinion, your domain is still affected by cross-boundary issues, as the Labrador current is not resolved in the domain but rather prescribed at the north ocean boundary. Please consider clarifying/re-writing. (This is not a criticism as all the regional models are subject to this, but more a request for clarification.)**

Our intent was to refer to issues occurring across fisheries management regions, Exclusive Economic Zones, and other geopolitical boundaries that our model's fairly large domain covers

(for example, a favorable environment for a fish shifting from U.S. to Canadian waters). However, we realize that our wording was not clear and thank the reviewer for pointing this out. We have tried to clarify by now stating that the model "covers a large "coast-wide" domain to address the prominent climate impacts expected to extend across fisheries management regions and other traditional geopolitical boundaries".

**7. Figure 1. Consider adding the names of some key regions in figure 1.a: e.g., the Northeast Channel, Cape Hatteras, Texas, Louisiana, Florida, Gulf of Mexico, Gulf of Saint Lawrence etc.**

We have added annotations for three potentially less-known features: the Northeast Channel, Cape Hatteras, and the LA-TX shelf.

[Figure]

**8. Lines 209-213. There is an emphasis on obtaining reliable solutions without applying restoring (e.g., surface salinity restoring). In my understanding, the simulation covers about 25-30 years (+ spin-up), and I was wondering if 25-30 years is a short time period in terms of the model developing a significant drift. Hence, I am not sure if the absence of restoring in the 25-hindcast run is indicative of the model's performance in longer simulations (e.g., climate projections), in terms of the emergence of significant drift; and if it will actually be beneficial to include a strategy for accounting for this drift in regional climate projections with your model. Maybe I have misunderstood something, but I suggest adding a brief discussion to clarify why it is expected that there will be no need for restoring to account for any drift in your regional ocean-only model under long-term simulations (longer than 25-30 years).**

This is a fair point. During development of the model, our experience was that errors that noticeably impacted the solution typically manifested as a drift away from reality within about 10 years or less. However, we acknowledge that it is possible that smaller drifts remain that are not noticeable in our 27-year simulation but could become relevant over climate-scale simulations. We think this is not a critical point (although we are pleased that the 27-year run does not appear to need restoring, MOM6 does have the ability to restore and it could be added in the future if longer runs show that it is necessary), so we have de-emphasized it in the text (deleting "it is important to note" and "deliberate") and added a note that "it will, however, be necessary to confirm that the model remains reliable over simulations longer than the 27-year run examined here". Finally, we will note here but not in the text that for work in progress we have run historical experiments over 60 years long without restoring and still obtained stable simulations.

**9. Lines 228-229. Please can you clarify how the rivers runoff salinity and temperature is treated (e.g., are you prescribing/assuming a constant 0 PSU salinity for river runoffs, or observed values?).**

We have added this sentence: "River discharge entered with zero salinity and a temperature equal to the surface temperature of the discharge grid cell".

**10. Line 404-405: I am not sure I understand what the feedback of biogeochemistry to tides will be in you model? (So I am not sure why there is expected to be even a negligible feedback). Please, can you clarify what this small feedback would involve (at least to the reply, as I was confused).**

We have deleted the mention of the expected negligible feedback on tides and now just state that biogeochemistry was not included for the tide estimation simulation. "Negligible" was an understatement here and we apologize for making it unclear. The only feedback from the biology to the ocean physics in the model is the ability of the simulated chlorophyll to modify light absorption and heating in the water. In principle this could affect the stratification and the generation of internal tides, but it is improbable that this would produce a meaningful difference in the tidal analysis.

**11. Lines 590-592 and Figure 7. To me, based on the 0.4 m shift in the colormap, it appears that the absolute dynamic topography and the model SSH have a difference/bias in magnitude. Is this maybe associated with the estimates of absolute dynamic topography and the geoid (I am not an expert on this so I am just curious about it)? I suggest, for clarity, to add the difference between model and observed SSH and the equivalent metrics as in the other figures (bias, RMSE, MedAR and Corr). This would help to better understand the magnitude and significance of the difference between the two datasets.**

Yes, this is likely associated with a difference between the geoid used in the reanalysis data and the satellite product. To reduce confusion, and because the reanalysis assimilates satellite data and has essentially the same long-term mean spatial pattern, we have removed the satellite

product from Figure 7. We added a note in the text that "The reanalysis assimilates satellite altimetry data and has a nearly identical long-term mean, aside from an apparent offset in the reference level". We also added the skill metrics comparing the model to reanalysis mean SSH to Figure 7.

[Figure]

**12. Figure 12. I suggest that you add the difference between model and OC-CCI satellite in the figure, as it is a little difficult to compare by eye where the model overestimates or underestimates the surface chlorophyll.**

In the revised version we have added the suggested panels that show the difference.

[Figure]

(a) DJF Model | (b) DJF OC-CCI | (c) DJF Model - OC-CCI

Bias: 0.01
RMSE: 0.21
MedAE: 0.12
Corr: 0.83

(d) MAM Model | (e) MAM OC-CCI | (f) MAM Model - OC-CCI

Bias: 0.04
RMSE: 0.24
MedAE: 0.13
Corr: 0.87

(g) JJA Model | (h) JJA OC-CCI | (i) JJA Model - OC-CCI

Bias: 0.00
RMSE: 0.26
MedAE: 0.16
Corr: 0.88

(j) SON Model | (k) SON OC-CCI | (l) SON Model - OC-CCI

Bias: -0.01
RMSE: 0.21
MedAE: 0.13
Corr: 0.89

0.25 0.5 1 2 4 8
Surface chl (mg / m$^3$)

0.25 0.5 1 2 4 8
Surface chl (mg / m$^3$)

-4 -2 -0.5 0 0.5 1 2 4
Difference. (mg m$^{-3}$)

**13. Figure 13. If I understood correctly, this is a zoom-in of figure 12 (maybe mention this in the Figure 13 caption). I suggest adding the bias, RMSE, MedAR and Corr metrics in the figure for the two different regions.**

Yes, this figure is a close-up of Figure 12. We have now stated this in the caption, and added the suggested skill metrics to the panels as well.

[Figure]

**14. Lines 727-728, Figure 21: It is a bit difficult to compare by eye the observations and model sea-ice concentration in Figure 21. Please consider adding the difference between observed and model in a third panel, as well as the associated metrics (bias, RMSE, MedAR and Corr).**

We have added panels for the difference and included the skill metrics.

[Figure]

**15. Lines 731-733. Please consider highlighting in the text that this is for the Gulf of St Lawrence, for example " … in sea ice coverage in the Gulf of St Lawrence, with correlation …).**

We have updated this to read: "The time series of monthly sea ice extent in the Gulf of St. Lawrence (Fig. 22) shows that the model captures nearly all of the year-to-year variability in sea ice coverage in the Gulf…"

**16. Typo, Lines 87-89: decadal time scale is repeated, maybe consider removing the "At decadal timescales" in the beginning.**

Fixed in the revised version.

---

## Author Comment (AC2)

**Summary:**

We thank the reviewer for providing a number of helpful suggestions on where clarification or additional information is needed. Below, we have restated each review comment in bold text followed by our reply.

**1: L31: The 2018 Brickman et al. paper documents and reveals the mechanism for these intrusions and should be referenced here:**

**Brickman, D., Hebert, D. and Wang, Z., 2018. Mechanism for the recent ocean warming events on the Scotian Shelf of eastern Canada. Continental Shelf Research, 156, pp.11-22. https://doi.org/10.1016/j.csr.2018.01.001**

We have added the suggested reference.

**2: L73: Regarding NAO effects on fisheries. The Fisher et al. 2008 paper provides an excellent example of the NAO effect on fish distributions in the model area. The authors may want to check the paper out and include a reference to it.**

**Fisher, J.A., Frank, K.T., Petrie, B., Leggett, W.C. and Shackell, N.L., 2008. Temporal dynamics within a contemporary latitudinal diversity gradient. Ecology letters, 11(9), pp.883-897.**

We appreciate this suggestion and have added the reference.

**3: L158 …163: Does the z* coord system have partial bottom cells? Authors should clarify the vertical resolution of the bottom cell.**

Yes, we edited the text to read "Vertically, the model uses a z∗ coordinate (a height coordinate that is rescaled with the free surface; Adcroft and Campin, 2004) with 75 layers and partial bottom cells"

**4: S2.3 Spinup and Hindcast: L332-342: I am not sure that I followed the spinup procedure, or perhaps I do not understand the logic. The authors describe a 10y spinup for the BGCM component (using a perpetual 1993? -- clarify) which is then used as the initial BGCM field for the main model run, which starts from rest in 1993 using the Glorys TS field. This confuses me. Because the BGCM is part of "main model" then the physics model must also be spun up. Why not use the 10y spinup to start the physics model as well? It is rare to not spinup a model, even if it is initialized from a 3D "spun up" TS field. Please clarify this procedure.**

We acknowledge that temperature and salinity are commonly spun up beforehand in regional simulations. In this case, however, we are initializing using a reliable ocean reanalysis with the same horizontal resolution as the regional model, and our experience during model

development was that initializing directly from this reanalysis produced more accurate simulations than trying to initialize from a spun up run. We did try several spin up methods, including repeating the first year of the simulation or repeating the first 10 years, and found that the initial conditions produced by these runs caused large and persistent errors immediately when beginning the main run.

We did use a short spin up run for the biogeochemistry, however, because the data we have to initialize the BGC from are primarily coarse-resolution climatologies and running a spin up allows the BGC component time to adjust from these less accurate and inconsistent initial conditions, at least at the surface.

To help clarify, we added a sentence to the text: "During model development, we found that spinning up the physics first, by either repeating the first year or the first 10 years, produced less reliable initial conditions than the high resolution GLORYS12 reanalysis and led to substantial errors in the beginning of the simulation."

The text also now states that the BGC spin up simulation "for 10 years using 1993--2002 time series of forcings described previously" to clarify that it is using the time series, not a repeating year.

**5: L364: conservatively interpolated; some details on this would be helpful**

We updated this to read "For spatial comparisons where the observed product had a resolution of 1/4° or coarser, the model data was interpolated onto the observed product grid with a first-order conservative method." For chlorophyll, the text now says it was "interpolated onto the NWA12 model grid with a method that preserved the geometric mean chlorophyll". The first-order conservative method conserves area averages, which is useful when regridding data from fine to coarse resolution. If a log transform is applied before using the method, and the results are transformed back afterwards, the geometric mean is conserved, which is useful for log-normal fields like chlorophyll.

**6: L384: "introduce a small bias"; further explanation needed.**

We revised this text to read: "On average this will introduce a small shallow bias in the model MLD relative to the de Boyer Montégut (2004) MLD when the model mixed layer threshold is near or above 10 m or the diurnal cycle of 0–2 m temperature is large; however, as we examine only mixed layers during winter when mixing is deeper and the diurnal cycle is weaker, the difference should be negligible."

Although not detailed in the text, to confirm this we calculated the two different definitions of the mixed layer depth for each cast with data above 2 m and below 40 m in the CTD database that we also used for the Northeast Channel metric. Averaged over all profiles (7,109)  from January–March, the average difference is 3.6 m shallower for the model definition of MLD. The

10th percentile of the difference is -11 m (i.e., in only 10% of profiles is the model definition more than 11 m shallower than the dBM definition).

**7: L385 … (and Fig 6), re GS position: There are a number of recent papers discussing changes in the GS position. For a slightly different analysis the authors should have a look at Wang et al. (2022) [Wang, Z., Yang, J., Johnson, C. and DeTracey, B., 2022. Changes in Deep Ocean Contribute to a "See‑Sawing" Gulf Stream Path. Geophysical Research Letters, 49(21), p.e2022GL100937. https://doi.org/10.1029/2022GL100937]**

We agree that there are quite a few papers on this topic, probably more than we can succinctly mention in this paragraph. We tried to cite papers that mentioned a connection to other changes in the region, such as warming or increased salinity. However, we agree the suggested citation is also useful and so we have modified the sentence in the introduction to read: "Northward shifts of the Gulf Stream have cut off the cool, southward Labrador Current and amplified warming in the region (Brickman et al., 2018; Gonçalves Neto et al., 2021; Seidov et al., 2021), although some studies have found contrasting long-term trends in the latitudinal position of the Gulf Stream (Wang et al., 2022)."

**8: L453, re EPUs (Figures 1b, 18): For researchers working on the SS/GSL/NL region, the SS EPU would not be considered part of the SS. This should be changed to eastern GoM (EGOM) as in Pontavice et al.**

We agree that this region is unfortunately named. However, for clarity and reproducibility we want to be consistent with the dataset used by NOAA Fisheries in their State of the Ecosystem reports (which we also use for observations; (https://noaa-edab.github.io/tech-doc/epu.html#epu), which calls this the Scotian Shelf-Eastern Gulf of Maine EPU. We have thus revised the text and Figure 18 to use the full name for the Scotian Shelf-Eastern Gulf of Maine EPU. We still abbreviate it as "SS" in Figure 1b, however, to be consistent with the abbreviation used in the EPU bottom temperature dataset.

**9: Results: 3.1**

**GS position (F6): F6a,b: lon/lat on these panels please**

Fixed in revised version.

**L593: "cross-shore" is a bit confusing; consider north-south or meridional**

Fixed in revised version.

**No need for Fig12. Fig13 shows the results better**

We have kept Figure 12 because it shows some other aspects of the chlorophyll (e.g. the subtropical gyre minimum and along the South American coast). Also, in response to Reviewer

1, we have added panels to this plot showing the difference between the model and OC-CCI, which make this figure more useful.

[Figure]

| (a) DJF Model | (b) DJF OC-CCI | (c) DJF Model - OC-CCI |
| (d) MAM Model | (e) MAM OC-CCI | (f) MAM Model - OC-CCI |
| (g) JJA Model | (h) JJA OC-CCI | (i) JJA Model - OC-CCI |
| (j) SON Model | (k) SON OC-CCI | (l) SON Model - OC-CCI |

(c) Bias: 0.01 RMSE: 0.21 MedAE: 0.12 Corr: 0.83

(f) Bias: 0.04 RMSE: 0.24 MedAE: 0.13 Corr: 0.87

(i) Bias: 0.00 RMSE: 0.26 MedAE: 0.16 Corr: 0.88

(l) Bias: -0.01 RMSE: 0.21 MedAE: 0.13 Corr: 0.89

Surface chl (mg / m$^3$)

Surface chl (mg / m$^3$)

Difference. (mg m$^{-3}$)

**L646: for clarity add the word "model" before "mesozooplankton".**

Fixed in revised version.

**10: S3.2**
**Deep Salinity: Fig19:**
**(a) TS fig. Please make colored symbols in the legend bigger so they can be seen.**

Fixed in revised version.

[Figure]

**(b,c) plots of %water masses (LSW & GSW): No mention of corrs in caption and it is not clear that the Glorys12 values are the model vs Glorys12 or Glorys12 vs one of the data series**

We added to the caption "Each correlation in parentheses gives the correlation between the model time series and the given observation dataset."

**Ice: F21,22: many models have problems with the advance, extent, and retreat of sea ice in the GSL (and on the NL shelf). This model does very well in this regard.**

We appreciate the comment and are also very pleased with how well the model simulates the sea ice.

**11. S3.3 Computational performance: Authors are commended on including this section. A couple of questions:**

**L771 (last sentence): Re scaling efficiency: The reason is not clear to me as wouldn't the (15x15 point / PE) restriction from the 2D BT solver still apply even when the BGCM component is implemented because the physics module still uses the BT solver? Please clarify.**

Yes, the BT solver would remain a bottleneck for the physics component of the model. However, with 40 additional tracers, the model with BGC spends a lot more time in the tracer routines. If these tracer routines continue to become faster as the number of PEs is increased, this would allow the model to continue to run faster with more PEs even though the barotropic solver is no longer becoming faster.

As a specific example, we compared runs from the 50x50 and 70x70 cases with biogeochemistry and looked at the average run times of individual steps and subroutines that MOM6 prints out at the end of the model run. The "Ocean barotropic mode stepping" step took a total of 2453 seconds in the 50x50 case and 2487 seconds for the 70x70 case; in other words, increasing the number of PEs from 50x50 had no impact on the time spent in the BT step. On the other hand, the "Ocean thermodynamics and tracers" step, which includes all of the biogeochemistry routines, took 15211 seconds for the 50x50 case and 10022 seconds for the 70x70 case, a speedup of about 1.5x and a meaningful reduction in the total run time. This speedup would not be nearly as prominent in a physics-only model with just temperature and salinity tracers.

We tried to clear this up in the text by changing it to:
"Furthermore, the scaling efficiency of this tracer-heavy model is better than seen in MOM6 models without coupled biogeochemistry where the computationally expensive 2-dimensional barotropic solver generally ceases to scale with fewer than 15x15 points per PE. The 40 additional tracers added by the BGC component result in more time spent in the tracer routines, which scale fairly well as the number of points per PE is reduced (and total PEs is increased) and produce a meaningful reduction in runtime even though the barotropic solver does not."

**L783: I do not understand this sentence, in particular the statement about reduced number of PEs needed by the longer thermodynamics timestep. I would have thought that the longer timestep requires fewer clock cycles, not fewer PEs. Perhaps my confusion is related to my comment above(?). In any case please clarify this**

We have deleted this sentence to reduce confusion. The sentence was referring to the computational resources needed to complete a run in a fixed amount of wall clock time. For example, if it was desired to complete a year-long simulation in exactly 12 hours, increasing the thermodynamics time step would allow the number of PEs to be decreased while maintaining the 12-hour run time. However, we realize this could be confusing and is not particularly important, so we have deleted this sentence.

**12: Discussion: L810: The Brickman et al. (2018) reference is relevant here as well.**

We have added the suggested reference.

**13: Conclusions: L866: I note that the authors mention a 0.5 corr as "commonly considered lower bound for useful prediction skill" although they do not provide a reference**

We added a citation to Murphy and Epstein 1989 ("Skill Scores and Correlation Coefficients in Model Verification"). We also changed "commonly considered" to "often considered" which in hindsight we think is more accurate since this is generally treated as a rule of thumb.